# THE KFIoU LOSS FOR ROTATED OBJECT DETECTION

**Xue Yang**[1], **Yue Zhou**[1], **Gefan Zhang**[1,2], **Jirui Yang**[3], **Wentao Wang**[1], **Junchi Yan**[1]*
**Xiaopeng Zhang**[4], **Qi Tian**[4]
[1]MoE Key Lab of Artificial Intelligence, Shanghai Jiao Tong University
[2]COWAROBOT Co. Ltd.  [3]University of Chinese Academy of Sciences  [4]Huawei Cloud
{yangxue-2019-sjtu,sjtu_zy,lizaozhouke,wwt117,yanjunchi}@sjtu.edu.cn
yangjirui123@gmail.com  {zhangxiaopeng12,tian.qi1}@huawei.com
Jittor Code:  https://github.com/Jittor/JDet
PyTorch Code:  https://github.com/open-mmlab/mmrotate
TensorFlow Code:  https://github.com/yangxue0827/RotationDetection

## ABSTRACT

Differing from the well-developed horizontal object detection area whereby the computing-friendly IoU based loss is readily adopted and well fits with the detection metrics, rotation detectors often involve a more complicated loss based on SkewIoU which is unfriendly to gradient-based training. In this paper, we propose an effective approximate SkewIoU loss based on Gaussian modeling and Gaussian product, which mainly consists of two items. The first term is a scale-insensitive center point loss, which is used to quickly narrow the distance between the center points of the two bounding boxes. In the distance-independent second term, the product of the Gaussian distributions is adopted to inherently mimic the mechanism of SkewIoU by its definition, and show its alignment with the SkewIoU loss at trend-level within a certain distance (i.e. within 9 pixels). This is in contrast to recent Gaussian modeling based rotation detectors e.g. GWD loss and KLD loss that involve a human-specified distribution distance metric which require additional hyperparameter tuning that vary across datasets and detectors. The resulting new loss called KFIoU loss is easier to implement and works better compared with exact SkewIoU loss, thanks to its full differentiability and ability to handle the non-overlapping cases. We further extend our technique to the 3-D case which also suffers from the same issues as 2-D. Extensive results on various datasets with different base detectors show the effectiveness of our approach.

## 1 INTRODUCTION

Rotated object detection is a relatively emerging but challenging area, due to the difficulties of locating the arbitrary-oriented objects and separating them effectively from the background, such as aerial images (Yang et al., 2018a; Ding et al., 2019; Yang et al., 2018b), scene text (Jiang et al., 2017; Zhou et al., 2017). Though considerable progresses have been recently made, for practical settings, there still exist challenges for rotating objects with large aspect ratio, dense distribution.

The Skew Intersection over Union (SkewIoU) between large aspect ratio objects is sensitive to the deviations of the object positions. This causes the negative impact of the inconsistency between metric (dominated by SkewIoU) and regression loss (e.g. $l_n$-norms), which is common in horizontal detection, and is further amplified in rotation detection. The red and orange arrows in Fig. 1 show the inconsistency between SkewIoU and Smooth L1 Loss. Specifically, when the angle deviation is fixed (red arrow), SkewIoU will decrease sharply as the aspect ratio increases, while the Smooth L1 loss is unchanged (mainly from the angle difference). Similarly, when SkewIoU does not change (orange arrow), Smooth L1 loss increases as the angle deviation increases.

---

*Correspondence author is Junchi Yan who is also affiliated with Shanghai AI Laboratory. The work was partly done when the first author Xue Yang was an intern at Huawei Cloud. The work was also in part supported by NSFC (62222607), Shanghai Municipal Science and Technology Major Project (2021SHZDZX0102).

Solution for inconsistency between the metric and regression loss has been extensively discussed in horizontal detection by using IoU loss and related variants, such as GIoU loss (Rezatofighi et al., 2019) and DIoU loss (Zheng et al., 2020b). However, the applications of these solutions to rotation detection are blocked because the analytical solution of the SkewIoU calculation process[1] is not easy to be provided due to the complexity of intersection between two rotated boxes (Zhou et al., 2019). Especially, there exist some custom operations (intersection of two edges and sorting the vertexes etc.) whose derivative functions have not been implemented in the existing deep learning frameworks (Abadi et al., 2016; Paszke et al., 2017; Hu et al., 2020). Besides, the calculation of SkewIoU is not differen-

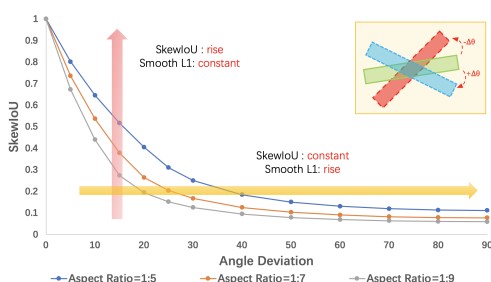

Figure 1: For rotation detection (Yang et al., 2021b), there is a notable inconsistency between the final detection metric i.e. mAP (largely depending on SkewIoU) and regression-based loss e.g. the popular Smooth L1. See Fig. 3(a) and Fig. 3(b) for more specific comparison.

tiable when there are more than eight intersection points between two bounding boxes, i.e. two boundary boxes are completely coincident, or one edge is coincident, which will lead to the failure to obtain very accurate prediction results. Thus, developing an easy-to-implement and fully differentiable approximate SkewIoU loss is meaningful and several works (Chen et al., 2020; Zheng et al., 2020a; Yang et al., 2021c;d) have been proposed.

This paper aims to find an easy-to-implement and better-performing alternative. We design an alternative to SkewIoU loss based on Gaussian product, named KFIoU loss[2], which can be easily implemented by the existing operations of the deep learning framework without the need for additional acceleration (e.g. C++/CUDA). Specifically, we convert the rotated bounding box into a Gaussian distribution, which can avoid the well-known boundary discontinuity and square-like problems (Yang et al., 2021c) in rotation detection. Then we use a center point loss to narrow the distance between the center of the two Gaussian distributions, follow by calculating the overlap area under the new position through the product of the Gaussian distributions. By calculating the error variance and comparing the final performance of different methods, we find trend-level alignment with the SkewIoU loss is critical for solving the inconsistency between metric and loss, and further improving the performance. Furthermore, compared to best-tuned Gaussian distance metric based methods, our proposed method achieves more competitive performance without hyperparameter tuning. **The highlights are as follows:**

1) For rotation detection, instead of exactly computing the SkewIoU loss which is tedious and unfriendly to differentiable learning, we propose our easy-to-implement approximate loss, named KFIoU loss, which works better since it is fully differentiable and able to handle the non-overlapping cases. It follows the protocol of Gaussian modeling for objects, yet innovatively uses Gaussian product to mimic SkewIoU's computing mechanism within a looser distance.

2) Compared to Gaussian-based losses (GWD loss, KLD loss) that try to approximate SkewIoU loss by specifying a distance which need extra hyperparameters tuning and metric selection that vary across datasets and detectors, our mechanism level simulation to SkewIoU is more interpretable and natural, and free from hyperparameter tuning.

3) We also show that KFIoU loss achieves the better trend-level alignment with SkewIoU loss within a certain distance than GWD loss and KLD loss, where the trend deviation is measured by our devised error variance. The effectiveness of such a trend-level alignment strategy is verified by comparing KFIoU loss with ideal SkewIoU loss. On extensive benchmarks (aerial images, scene texts, face), our approach also outperforms other best-tuned SOTA alternatives.

4) We further extend the Gaussian modeling and KFIoU loss from 2-D to 3-D rotation detection, with notable improvement compared with baselines. To our best knowledge, this is the first 3-D rotation detector based on Gaussian modeling which also verifies its effectiveness, which is in

---

[1]See an open-source version with thousands of lines of code for implementing the loss at https://github.com/open-mmlab/mmcv/pull/1854, while our new loss only costs tens of lines of code.

[2]we term our loss as KFIoU as the product of Gaussian is an important step in Kalman filtering.

contrast to (Yang et al., 2021c;d; 2022) focusing on 2-D rotation detection. The source code is available at TensoFlow (Abadi et al., 2016)-based AlphaRotate (Yang et al., 2021e), PyTorch (Paszke et al., 2017)-based MMRotate (Zhou et al., 2022) and Jittor (Hu et al., 2020)-based JDet.

## 2 RELATED WORK

**Rotated Object Detection.** Rotated object detection is an emerging direction, which attempts to extend classical horizontal detectors (Girshick, 2015; Ren et al., 2015; Lin et al., 2017a;b) to the rotation case by adopting the rotated bounding boxes. Aerial images and scene text are popular application scenarios of rotation detector. For aerial images, objects are often arbitrary-oriented and dense-distributed with large aspect ratios. To this end, ICN (Azimi et al., 2018), ROI-Transformer (Ding et al., 2019), SCRDet (Yang et al., 2019), Mask OBB (Wang et al., 2019), Gliding Vertex (Xu et al., 2020), ReDet (Han et al., 2021b) are two-stage mainstreamed approaches whose pipeline is inherited from Faster RCNN (Ren et al., 2015), while DRN (Pan et al., 2020), DAL (Ming et al., 2021b), R$^3$Det (Yang et al., 2021b), RSDet (Qian et al., 2021a;b) and S$^2$A-Net (Han et al., 2021a) are based on single-stage methods for faster detection speed. For scene text detection, RRPN (Ma et al., 2018) employs rotated RPN to generate rotated proposals and further perform rotated bounding box regression. TextBoxes++ (Liao et al., 2018a) adopts vertex regression on SSD (Liu et al., 2016). RRD (Liao et al., 2018b) improves TextBoxes++ by decoupling classification and bounding box regression on rotation-invariant and rotation sensitive features, respectively. The regression loss of the above algorithms is rarely SkewIoU loss due to the complexity of implementing SkewIoU.

**Variants of IoU-based Loss.** The inconsistency between metric and regression loss is a common issue for both horizontal detection and rotation detection. Solution for this inconsistency has been extensively discussed in horizontal detection by using IoU related loss. For instance, Unitbox (Yu et al., 2016) proposes an IoU loss which regresses the four bounds of a predicted box as a whole unit. More works (Rezatofighi et al., 2019; Zheng et al., 2020b) extend the idea of Unitbox by introducing GIoU (Rezatofighi et al., 2019) and DIoU (Zheng et al., 2020b) for bounding box regression. However, their applications to rotation detection are blocked due to the hard-to-implement SkewIoU. Recently, some approximate methods for SkewIoU loss have been proposed. **Box/Polygon based:** SCRDet (Yang et al., 2019) propose IoU-Smooth L1, which partly circumvents the need for SkewIoU loss with gradient backpropagation by combining IoU and Smooth L1 loss. To tackle the uncertainty of convex caused by rotation, the work (Zheng et al., 2020a) proposes a projection operation to estimate the intersection area for both 2-D/3-D object detection. PolarMask (Xie et al., 2020) proposes Polar IoU loss that can largely ease the optimization and considerably improve the accuracy. CFA (Guo et al., 2021) proposes convex hull based CIoU loss for optimization of point based detectors. **Pixel based:** PIoU (Chen et al., 2020) calculates the SkewIoU directly by accumulating the contribution of interior overlapping pixels. **Gaussian based:** GWD (Yang et al., 2021c) and KLD (Yang et al., 2021d) simulate SkewIoU by Gaussian distance measurement.

## 3

This section presents the preliminary according to (Yang et al., 2021c), for how to convert an arbitrary-oriented 2-D/3-D bounding box to a Gaussian distribution $\mathcal{G}(\mu, \mathbf{\Sigma})$.

$$\mathbf{\Sigma} = \mathbf{R}\mathbf{\Lambda}\mathbf{R}^\top, \ \mu = (x, y, (z))^\top \tag{1}$$

where $\mathbf{R}$ represents the rotation matrix, and $\mathbf{\Lambda}$ represents the diagonal matrix of eigenvalues.

Take 3-D object $\mathcal{B}_{3d}(x, y, z, w, h, l, \theta)$ as an example:

$$\mathbf{R} = \begin{pmatrix} \cos\theta & -\sin\theta & 0 \\ \sin\theta & \cos\theta & 0 \\ 0 & 0 & 1 \end{pmatrix}, \ \mathbf{\Lambda} = \begin{pmatrix} \frac{w^2}{4} & 0 & 0 \\ 0 & \frac{h^2}{4} & 0 \\ 0 & 0 & \frac{l^2}{4} \end{pmatrix} \tag{2}$$

It is worth noting that the recent works GWD loss (Yang et al., 2021c) and KLD loss (Yang et al., 2021d) also belong to the Gaussian modeling based. Compared with our work, their difference is that they use the nonlinear transformation of distribution distance to approximate SkewIoU loss. In this process, additional hyperparameters are introduced. Since Gaussian modeling has the natural

Table 1: Comparison of the properties and performance of different regression losses. Base model is RetinaNet. **BC** and **HP** denote Boundary Continuity and Hyperparameter. [†] indicates that the first term of KLD is taken as the center point loss, i.e. $L_c(\boldsymbol{\mu}_1, \boldsymbol{\mu}_2, \boldsymbol{\Sigma}_1)$.

| Loss | Representation | Implement | BC | Consistency | HP | EVar↓ | DOTA-v1.0 | DOTA-v1.5 | DOTA-v2.0 |
|---|---|---|---|---|---|---|---|---|---|
| Smooth L1 | bbox | easy | × | × | ✓ ($\sigma$) | 0.073201718 | 64.17 | 56.10 | 43.06 |
| plain SkewIoU | bbox | hard | ✓ | ✓ | × | - | 68.27 | 59.01 | 45.87 |
| GWD | Gaussian | easy | ✓ | × | ✓ ($\tau, f$) | 0.019041297 | 68.93 | 60.03 | 46.65 |
| KLD | Gaussian | easy | ✓ | ✓ | ✓ ($\tau, f$) | 0.007653582 | 71.28 | 62.50 | 47.69 |
| KFIoU (ours) | Gaussian | easy | ✓ | ✓ | × | 0.002348353 | 70.64 | 62.71 | 48.04 |
| KFIoU[†] (ours) | Gaussian | easy | ✓ | ✓ | × | **0.002264243** | **71.60** | **63.75** | **48.94** |

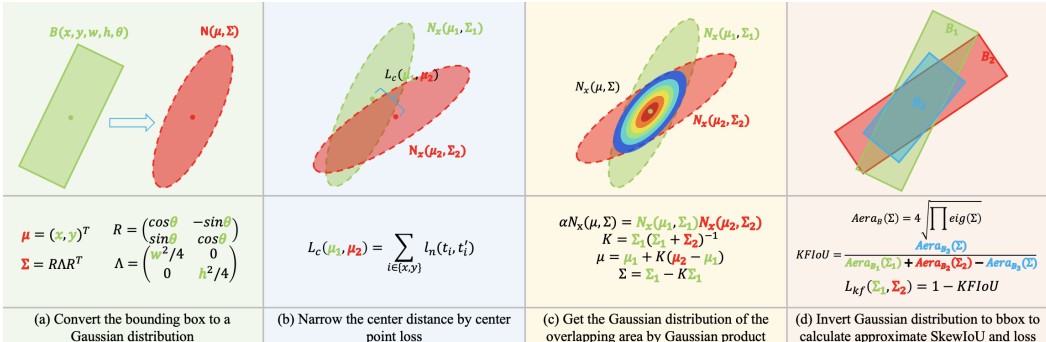

(a) Convert the bounding box to a Gaussian distribution    (b) Narrow the center distance by center point loss    (c) Get the Gaussian distribution of the overlapping area by Gaussian product    (d) Invert Gaussian distribution to bbox to calculate approximate SkewIoU and loss

Figure 2: SkewIoU loss approximation process in two-dimensional space based on Gaussian product. Compared with GWD loss (Yang et al., 2021c) and KLD loss (Yang et al., 2021d), our approach follows the calculation process of SkewIoU without introducing additional hyperparameters. We believe such a design is more mathematically rigorous and more in line with SkewIoU loss.

advantages of being immune to boundary discontinuity and square-like problems, in this paper, we will take another perspective to approximate the SkewIoU loss to better train the detector without any extra hyperparameter, which can be more in line with SkewIoU calculation. Tab. 1 shows the comparison of properties between different losses. It should be noted that the results presented in our experiments of GWD loss and KLD loss are obtained by best-tuned hyperparameters in DOTA, but not optimal in others.

## 4 PROPOSED METHOD

In this section, we present our main approach. Fig. 2 shows the approximate process of SkewIoU loss in two-dimensional space based on Gaussian product. Briefly, we first convert the bounding box to a Gaussian distribution as discussed in Sec. 3, and move the center points of the two Gaussian distributions to make them close. Then, the distribution function of the overlapping area is obtained by Gaussian product. Finally, the obtained distribution function is inverted into a rotated bounding box to calculate the overlapping area and the SkewIoU and loss.

### 4.1 SKEWIOU BASED ON GAUSSIAN PRODUCT

First of all, we can easily calculate the volume of the corresponding rotating box based on its covariance ($\mathcal{V}_\mathcal{B}(\boldsymbol{\Sigma})$), when we obtain a new Gaussian distribution:

$$\mathcal{V}_\mathcal{B}(\boldsymbol{\Sigma}) = 2^n \sqrt{\prod eig(\boldsymbol{\Sigma})} = 2^n \cdot |\boldsymbol{\Sigma}^{\frac{1}{2}}| = 2^n \cdot |\boldsymbol{\Sigma}|^{\frac{1}{2}} \tag{3}$$

where $n$ denotes the number of dimensions.

To obtain the final SkewIoU, calculating the area of overlap is critical. For two Gaussian distributions, $\mathcal{N}_\mathbf{x}(\mu_1, \boldsymbol{\Sigma}_1)$ and $\mathcal{N}_\mathbf{x}(\mu_2, \boldsymbol{\Sigma}_2)$, we use the product of the Gaussian distributions to get the distribution function of the overlapping area:

$$\alpha \mathcal{N}_\mathbf{x}(\mu, \boldsymbol{\Sigma}) = \mathcal{N}_\mathbf{x}(\mu_1, \boldsymbol{\Sigma}_1) \mathcal{N}_\mathbf{x}(\mu_2, \boldsymbol{\Sigma}_2) \tag{4}$$

Note here $\alpha$ is written by:

$$\alpha = \mathcal{N}_{\mu_1}(\mu_2, \boldsymbol{\Sigma}_1 + \boldsymbol{\Sigma}_2) \tag{5}$$

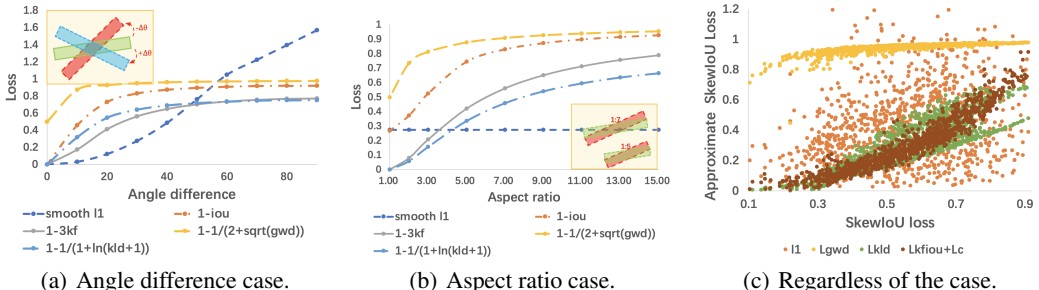

(a) Angle difference case.     (b) Aspect ratio case.     (c) Regardless of the case.

Figure 3: Behavior comparison of different losses in different cases. (a) depicts the relation between angle difference and loss functions. (b) shows the changes of the five loss under different aspect ratio condition. (c) gives scatter plot between approximate losses and SkewIoU loss, 1,000 examples regardless of the case by randomly generating box pairs with the close centers (within 5 pixels).

where $\mu = \mu_1 + \mathbf{K}(\mu_2 - \mu_1)$, $\Sigma = \Sigma_1 - \mathbf{K}\Sigma_1$, and $\mathbf{K}$ is the Kalman gain, $\mathbf{K} = \Sigma_1(\Sigma_1 + \Sigma_2)^{-1}$.

We observe that $\Sigma$ is only related to the covariance ($\Sigma_1$ and $\Sigma_2$) of the given two Gaussian distributions, which means that no matter how the two Gaussian distributions move, as long as the covariance is fixed, the area calculated by Eq. 3 will not change (distance-independent). This is obviously not in line with intuition: the overlapping area should be reduced when the two Gaussian distributions are far away. The main reason is $\alpha \mathcal{N}_{\mathbf{x}}(\mu, \Sigma)$ is not a standard Gaussian distribution (probability sum is not 1), we cannot directly use $\Sigma$ to calculate the area of the current overlap by Eq. 3 without considering $\alpha$. Eq. 5 shows that $\alpha$ is related to the distance between the center points ($\mu_1 - \mu_2$) of the two Gaussian distributions. Based on the above findings, we can first use a center point loss $L_c$ to narrow the distance between the center of the two Gaussian distributions. In this way, $\alpha$ can be approximated as a constant, and the introduction of the $L_c$ also allows the entire loss to continue to optimize the detector in non-overlapping cases. Then, calculate the overlap area under the new position by Eq. 3. According to Fig. 2, overlap area is calculated as follows:

$$\text{KFIoU} = \frac{\mathcal{V}_{\mathcal{B}_3}(\Sigma)}{\mathcal{V}_{\mathcal{B}_1}(\Sigma_1) + \mathcal{V}_{\mathcal{B}_2}(\Sigma_2) - \mathcal{V}_{\mathcal{B}_3}(\Sigma)} \tag{6}$$

where $\mathcal{B}_1$, $\mathcal{B}_2$ and $\mathcal{B}_3$ refer to the three different bounding boxes shown in the right part of Fig. 2.

In the appendix, we prove that the upper bounds of KFIoU in n-dimensional space is $\frac{1}{2^{\frac{n}{2}+1}-1}$. For 2-D/3-D detection, the upper bounds are $\frac{1}{3}$ and $\frac{1}{\sqrt{32}-1}$ respectively when $n = 2$ and $n = 3$. We can easily stretch the range of KFIoU to $[0, 1]$ by linear transformation according to the upper bound, and then compare it with IoU for consistency.

Fig. 3(a)-3(b) show the curves of five loss forms for two bounding boxes with the same center in different cases. Note that we have expanded KFIoU by 3 times so that its value range is $[0, 1]$ like SkewIoU. Fig. 3(a) depicts the relation between angle difference and loss functions. Though they all bear monotonicity, obviously the Smooth L1 loss curve is more distinctive. Fig. 3(b) shows the changes of the five loss under different aspect ratio conditions. It can be seen that the Smooth L1 loss of the two bounding boxes are constant (mainly from the angle difference), but other losses will change drastically as the aspect ratio varies. Regardless of the case in Fig. 3(c), KFIoU loss can maintain the best trend-level alignment with the SkewIoU loss within 5 pixels deaviariation. This conclusion still holds at 9 pixels, which is already quite a distance, especially for aerial image.

To further explore the behavior of different approximate SkewIoU losses, we design the metrics of error mean (EMean) and error variance (EVar) as follows:

$$\text{EMean} = \frac{1}{N} \sum_{i=1}^{N} (\text{SkewIoU}_{plain} - \text{SkewIoU}_{app}), \quad \text{EVar} = \frac{1}{N} \sum_{i=1}^{N} (\text{SkewIoU}_{app} - \text{EMean})^2 \tag{7}$$

where EVar measures the trend-level consistency between the designed loss and the SkewIoU loss.

Tab. 1 calculates the EVar of different losses in Fig. 3(c). In general, $\text{EVar}_{L_{kfiou}+L_c} < \text{EVar}_{L_{kld}} < \text{EVar}_{L_{gwd}} < \text{EVar}_{L_1}$. In our analysis, this is probably due to the fundamental inconsistency between

the distribution distance as used in GWD/KLD and the definition of similarity in SkewIoU. Moreover, for GWD such inconsistency is more pronouced, because it has no scale invariance under the same IoU, and a case with a larger scale will get a larger loss value, it can greatly magnify its trend inconsistency with SkewIoU loss. The results in Tab. 1 also verifies our analysis. In contrast, the calculation process of KFIoU loss is essentially the calculation of the overlap rate, so it does not require hyperparameters and can maintain a high trend-level consistency with SkewIoU loss.

Combined with the corresponding performance on three datasets, smaller EVars tend to have better performance in a general level. When EVar is small enough, which implies sufficient consistency, the performance difference of different methods (e.g. KLD loss and KFIoU loss) is close. Therefore, we come to the conclusion that the key to maintaining the consistency between metric and regression loss lies in the trend-level consistency between approximate and exact SkewIoU loss rather than value-level consistency. The reason why the Gaussian-based losses (e.g. KFIoU loss, KLD loss, GWD loss) outperform the plain SkewIoU loss is due to the advanced parameter optimization mechanism, effective measurement for non-overlapping cases, and complete derivation. However, the introduction of hyperparameters makes KLD loss and GWD loss less stable than KFIoU loss in terms of Evar and performance. Compared with GWD and KLD, which use the distribution distance to approximate SkewIoU, KFIoU is physically more reasonable (in line with the calculation process of SkewIoU) and simpler, as well as empirically more effective than best-tuned GWD and KLD. In addition, KFIoU implementation is much simpler than plain SkewIoU and can be easily implemented by the existing operations of the deep learning framework.

## 4.2 THE PROPOSED KFIoU LOSS

We take 2-D object detection as the main example for notation brevity, though our experiments further cover the 3-D case. We use the one-stage detector RetinaNet (Lin et al., 2017b) as the baseline. Rotated rectangle is represented by five parameters $(x, y, w, h, \theta)$. First, we shall clarify that the network has not changed the output of the original regression branch, that is, it is not directly predicting the parameters of the Gaussian distribution. The whole training process of detector is summarized as follows: i) predict offset $(t_x^*, t_y^*, t_w^*, t_h^*, t_\theta^*)$; ii) decode prediction box; iii) convert prediction box and target ground-truth into Gaussian distribution; iv) calculate $L_c$ and $L_{kf}$ of two Gaussian distributions. Therefore, the inference time remains unchanged. The regression equation of $(x, y, w, h)$ is as follows:

$$
\begin{aligned}
t_x &= (x - x_a)/w_a, \quad t_y = (y - y_a)/h_a, \quad t_w = \log(w/w_a), \quad t_h = \log(h/h_a) \\
t_x^* &= (x^* - x_a)/w_a, \quad t_y^* = (y^* - y_a)/h_a, \quad t_w^* = \log(w^*/w_a), \quad t_h^* = \log(h^*/h_a)
\end{aligned}
\tag{8}
$$

where $x, y, w, h$ denote the box's center coordinates, width and height, respectively. $x, x_a, x^*$ are for ground-truth box, anchor box, and predicted box (likewise for $y, w, h$).

For the regression of $\theta$, we use two forms as the baselines:

i) Direct regression, marked as **Reg. ($\Delta\theta$)**. The model directly predicts the angle offset $t_\theta^*$:

$$
t_\theta = (\theta - \theta_a) \cdot \pi/180, \quad t_\theta^* = (\theta^* - \theta_a) \cdot \pi/180
\tag{9}
$$

ii) Indirect regression, marked as **Reg.$^*$ ($\sin\theta$, $\cos\theta$)**. The model predicts two vectors ($t_{\sin\theta}^*$ and $t_{\cos\theta}^*$) to match the two targets from the ground truth ($t_{\sin\theta}$ and $t_{\cos\theta}$):

$$
t_{\sin\theta} = \sin(\theta \cdot \pi/180), \quad t_{\cos\theta} = \cos(\theta \cdot \pi/180), \quad t_{\sin\theta}^* = \sin(\theta^* \cdot \pi/180), \quad t_{\cos\theta}^* = \cos(\theta^* \cdot \pi/180)
\tag{10}
$$

To ensure that $t_{\sin\theta}^{*2} + t_{\cos\theta}^{*2} = 1$ is satisfied, we will perform the following normalization processing:

$$
t_{\sin\theta}^* = \frac{t_{\sin\theta}^*}{\sqrt{t_{\sin\theta}^{*2} + t_{\cos\theta}^{*2}}}, \quad t_{\cos\theta}^* = \frac{t_{\cos\theta}^*}{\sqrt{t_{\sin\theta}^{*2} + t_{\cos\theta}^{*2}}}
\tag{11}
$$

The multi-task loss is:

$$
L_{total} = \lambda_1 \sum_{n=1}^{N_{pos}} L_{reg}\left(\mathcal{G}(b_n), \mathcal{G}(gt_n)\right) + \frac{\lambda_2}{N} \sum_{n=1}^{N} L_{cls}(p_n, t_n)
\tag{12}
$$

where $N$ and $N_{pos}$ indicates the number of all anchors and that of positive anchors. $b_n$ denotes the $n$-th predicted bounding box, $gt_n$ is the $n$-th target ground-truth. $\mathcal{G}(\cdot)$ is Gaussian transfer function.

Table 2: Ablation study on various 2-D datasets with different base detectors. 'R', 'F' and 'G' indicate random rotation, flipping, and graying. $^\dagger$ indicates that the first term of KLD is taken as the center point loss, i.e. $L_c(\boldsymbol{\mu}_1, \boldsymbol{\mu}_2, \boldsymbol{\Sigma}_1)$. Base detector is RetinaNet.

| Dataset | Data Aug. | Reg. Loss | Hmean/AP$_{50}$ | Hmean/AP$_{60}$ | Hmean/AP$_{75}$ | Hmean/AP$_{85}$ | Hmean/AP$_{50:95}$ |
|---|---|---|---|---|---|---|---|
| HRSC2016 | R+F+G | Smooth L1 | 84.28 | 74.74 | 48.42 | 12.56 | 47.76 |
| | | KFIoU | **84.41 (+0.13)** | **82.23 (+7.49)** | **58.32 (+9.90)** | **18.34 (+5.78)** | **51.29 (+3.53)** |
| MSRA-TD500 | R+F | Smooth L1 | 70.98 | 62.42 | 36.73 | 12.56 | 37.89 |
| | | KFIoU | **76.30 (+5.32)** | **69.84 (+7.42)** | **47.58 (+10.85)** | **19.21 (+6.65)** | **44.96 (+7.07)** |
| ICDAR2015 | | Smooth L1 | 69.78 | 64.15 | 36.97 | 8.71 | 37.73 |
| | | KFIoU | **75.90 (+6.12)** | **69.28 (+5.13)** | **40.03 (+3.06)** | **9.18 (+0.47)** | **41.17 (+3.44)** |
| FDDB | F | Smooth L1 | 95.92 | 87.50 | 55.81 | 12.67 | 52.77 |
| | | KFIoU | **97.25 (+1.33)** | **94.89 (+7.39)** | **77.38 (+21.57)** | **25.62 (+12.93)** | **63.25 (+10.48)** |
| DOTA-v1.0 | | Smooth L1 | 65.00 | 57.84 | 33.68 | 11.39 | 35.16 |
| | | KFIoU | 67.68 (+2.68) | 62.18 (+4.34) | 37.30 (+3.62) | **14.21 (+2.82)** | 38.51 (+3.35) |
| | | KFIoU$^\dagger$ | **68.23 (+3.23)** | **63.23 (+5.39)** | **38.34 (+4.66)** | 13.72 (+2.33) | **38.80 (+3.64)** |

$t_n$ represents the label of the $n$-th object, $p_n$ is the $n$-th probability distribution of classes calculated by sigmoid function. $\lambda_1$, $\lambda_2$ control the trade-off and are set to $\{0.01, 1\}$. The classification loss $L_{cls}$ is set as the focal loss (Lin et al., 2017b). The regression loss is set by $L_{reg} = L_c + L_{kf}$, where

$$L_{kf}(\boldsymbol{\Sigma}_1, \boldsymbol{\Sigma}_2) = e^{1-\text{KFIoU}} - 1 \tag{13}$$

See more ablation experiments on the functional form of $L_{kf}(\boldsymbol{\Sigma}_1, \boldsymbol{\Sigma}_2)$ in the Appendix. For center point loss $L_c$, this paper provides two different forms:

**1)** The loss adopted in Faster RCNN (Lin et al., 2017a) (default): $L_c(t, t^*) = \sum_{i \in (x,y)} l_n(t_i, t_i^*)$.

**2)** The first term of KLD (Yang et al., 2021d) (advanced), which has an advanced center point optimization mechanism: $L_c(\boldsymbol{\mu}_1, \boldsymbol{\mu}_2, \boldsymbol{\Sigma}_1) = \ln\left((\boldsymbol{\mu}_2 - \boldsymbol{\mu}_1)^\top \boldsymbol{\Sigma}_1^{-1}(\boldsymbol{\mu}_2 - \boldsymbol{\mu}_1) + 1\right)$.

## 5 EXPERIMENTS

### 5.1 DATASETS AND IMPLEMENTATION DETAILS

**Aerial image dataset: DOTA** (Xia et al., 2018) is one of the largest datasets for oriented object detection in aerial images with three released versions: DOTA-v1.0, DOTA-v1.5 and DOTA-v2.0. DOTA-v1.0 contains 15 common categories, 2,806 images and 188,282 instances. DOTA-v1.5 uses the same images as DOTA-v1.0, but extremely small instances (less than 10 pixels) are also annotated. Moreover, a new category, containing 402,089 instances in total is added in this version. While DOTA-v2.0 contains 18 common categories, 11,268 images and 1,793,658 instances. We divide the images into $600 \times 600$ subimages with an overlap of 150 pixels and scale it to $800 \times 800$. **HRSC2016** (Liu et al., 2017) contains images from two scenarios with ships on sea and close inshore. The training, validation and test set include 436, 181 and 444 images.

**Scene text dataset: ICDAR2015** (Karatzas et al., 2015) includes 1,000 training images and 500 testing images. **MSRA-TD500** (Yao et al., 2012) has 300 training images and 200 testing images. They are popular for oriented scene text detection and spotting.

**Face dataset: FDDB** (Jain & Learned-Miller, 2010) is a dataset designed for unconstrained face detection, in which faces have a wide variability of face scales, poses, and appearance. This dataset contains annotations for 5,171 faces in a set of 2,845 images. We manually use 70% as the training set and the rest as the validation set.

We use AlphaRotate (Yang et al., 2021e) for main implementation and experiment, where many advanced detectors are integrated. Experiments are performed on a server with GeForce RTX 3090 Ti and 24G memory. Experiments are initialized by ResNet50 (He et al., 2016) by default unless otherwise specified. We perform experiments on two aerial benchmarks, two scene text benchmarks and one face benchmark to verify the generality of our techniques. Weight decay and momentum are set 0.0001 and 0.9, respectively. We employ MomentumOptimizer over 4 GPUs with a total of 4 images per mini-batch (1 image per GPU). All the used datasets are trained by 20 epochs, and learning rate is reduced tenfold at 12 epochs and 16 epochs, respectively. The initial learning rate is 1e-3. The number of image iterations per epoch for DOTA-v1.0, DOTA-v1.5, DOTA-v2.0, HRSC2016, ICDAR2015, MSRA-TD500 and FDDB are 54k, 64k, 80k, 10k, 10k, 5k and 4k respectively, and doubled if data augmentation (e.g. random graying and rotation) or multi-scale training are enabled.

Table 3: Results on KITTI val split 3D detection and BEV Detection.

| Method | mAP | 3D Detection Mod. | | | mAP | BEV Detection Mod. | | |
|---|---|---|---|---|---|---|---|---|
| PointPillars | 64.28 | 78.90 | 50.96 | 62.99 | 70.10 | 88.08 | 55.51 | 66.69 |
| +GWD | 65.50 | 78.57 | 55.19 | 62.74 | 71.48 | 88.30 | 58.49 | 67.66 |
| +KLD | 66.19 | 80.36 | 52.94 | 65.27 | 71.18 | 88.11 | 57.26 | 68.19 |
| +KFIoU | **66.71** | 80.19 | 54.94 | 65.00 | **72.08** | 89.90 | 57.81 | 68.55 |

**KITTI** (Geiger et al., 2012) contains 7,481 training and 7,518 testing samples for 3-D object detection. The training samples are generally divided into the train split (3,712 samples) and the val split (3,769 samples). The evaluation is classified into Easy, Moderate or Hard according to the object size, occlusion and truncation. All results are evaluated by the mean average precision with a rotated IoU threshold 0.7 for cars and 0.5 for pedestrian and cyclists. To evaluate the model's performance on KITTI val split, we train our model on the train set and report the results on the val set.

We use PointPillar (Lang et al., 2019) implemented in MMDetection3D (Contributors, 2020) as the baseline, and the training schedule inherited from SECOND (Yan et al., 2018): ADAM optimizer with a cosine-shaped cyclic learning rate scheduler that spans 160 epochs. The learning rate starts from 1e-4 and reaches 1e-3 at the 60th epoch, and then goes down gradually to 1e-7 finally. In the development phase, the experiments are conducted with a single model for 3-class joint detection.

## 5.2 ABLATION STUDY AND FURTHER COMPARISON

**Ablation study on different center point losses.** Tab. 1 compares the two different center point losses proposed in Sec. 4.2 on three versions of DOTA datasets. Even with the most commonly used $L_c(t, t^*)$, KFIoU loss achieves competitive performance, significantly better than GWD loss and comparable to KLD loss. For a fairer comparison, after adopting the same center point loss term as KLD loss $L_c(\boldsymbol{\mu}_1, \boldsymbol{\mu}_2, \boldsymbol{\Sigma}_1)$, the performance of KFIoU loss is further improved, which is better than KLD loss thanks to a better center point optimization mechanism.

**Ablation study on various 2-D datasets with different detectors.** Tab. 2 compares Smooth L1 loss and KFIoU loss by indicators with different IoU thresholds. For HRSC2016 containing a large number of ships with large aspect ratios, KFIoU loss has a **9.90%** improvement over Smooth L1 on $AP_{75}$. For the scene text datasets MSRA-TD500 and ICDAR2015, KFIoU achieves **7.07%** and **3.44%** improvements on $Hmean_{50:95}$, reaching **44.96%** and **41.17%** respectively. The same conclusion can be reached on FDDB and DOTA-v1.0 datasets.

**Ablation study of KFIoU loss on 3-D detection.** We generalize the KFIoU loss from 2-D to 3-D, with results in Tab. 3. It involves 3-D detection and BEV detection on KITTI val split, and

Table 4: Accuracy (%) comparison on DOTA. The bold red and blue indicate the top two performances. $D_{oc}$ and $D_{le}$ denotes OpenCV Definition ($\theta \in [-90°, 0°)$) and Long Edge Definition ($\theta \in [-90°, 90°)$) of RBox. 'H' and 'R' denote the horizontal and rotating anchors, respectively. $^\dagger$ indicates that the first term of KLD is taken as the center point loss, i.e. $L_c(\boldsymbol{\mu}_1, \boldsymbol{\mu}_2, \boldsymbol{\Sigma}_1)$.

| Method | Box Def. | DOTA-v1.0 | DOTA-v1.5 | DOTA-v2.0 |
|---|---|---|---|---|
| RetinaNet-H (Reg.) (2017b) | $D_{oc}$ | 65.73 | 58.87 | 44.16 |
| RetinaNet-H (Reg.) (2017b) | $D_{le}$ | 64.17 | 56.10 | 43.06 |
| RetinaNet-H (Reg.*) (2017b) | $D_{le}$ | 65.78 | 57.17 | 43.92 |
| RetinaNet-R (Reg.) (2017b) | $D_{oc}$ | 67.25 | 56.50 | 42.04 |
| PIoU (2020) | $D_{oc}$ | 65.85 | 57.65 | 45.23 |
| IoU-Smooth L1 (2019) | $D_{oc}$ | 66.99 | 59.16 | 46.31 |
| Modulated Loss (2021a) | $D_{oc}$ | 66.05 | 57.75 | 45.17 |
| Modulated Loss (2021a) | Quad. | 67.20 | 61.42 | 46.71 |
| RIL (2021a) | Quad. | 66.06 | 58.91 | 45.35 |
| CSL (2020) | $D_{le}$ | 67.38 | 58.55 | 43.34 |
| DCL (BCL) (2021a) | $D_{le}$ | 67.39 | 59.38 | 45.46 |
| plain SkewIoU (2019) | $D_{oc}$ | 68.27 | 59.01 | 45.87 |
| GWD (2021c) | $D_{oc}$ | 68.93 | 60.03 | 46.65 |
| KLD (2021d) | $D_{oc}$ | 71.28 | 62.50 | 47.69 |
| KFIoU (Ours) | $D_{oc}$ | 70.64 | 62.71 | 48.04 |
| KFIoU$^\dagger$ (Ours) | $D_{oc}$ | 71.60 | 63.75 | 48.94 |

significant performance improvements are also achieved. On the moderate level of 3-D detection and BEV detection, KFIoU loss improves PointPillars by **2.43%** and **1.98%**, respectively.

**Comparison with peer methods.** Methods in Tab. 4 are based on the same baseline RetinaNet, and initialized by ResNet50 (He et al., 2016) without using data augmentation and multi-scale training/testing. They are trained/tested under the same environment and hyperparameters. These methods are all published solutions to the boundary discontinuity in rotation detection.

First, we conduct ablation experiments on anchor form, rotated bounding box definition form, and angle regression form based on RetinaNet. Rotating anchors provides accurate prior, which makes

Table 5: AP of different objects on DOTA-v1.0. **Red** and **blue**: top two performances.

| | Method | Backbone | PL | BD | BR | GTF | SV | LV | SH | TC | BC | ST | SBF | RA | HA | SP | HC | mAP$_{50}$ |
|---|---|---|---|---|---|---|---|---|---|---|---|---|---|---|---|---|---|---|
| **Single-stage** | PIoU (2020) | DLA-34 | 80.90 | 69.70 | 24.10 | 60.20 | 38.30 | 64.40 | 64.80 | 90.90 | 77.20 | 70.40 | 46.50 | 37.10 | 57.10 | 61.90 | 64.00 | 60.50 |
| | O²-DNet (2020a) | H-104 | 89.31 | 82.14 | 47.33 | 61.21 | 71.32 | 74.03 | 78.62 | 90.76 | 82.23 | 81.36 | 60.93 | 60.17 | 58.21 | 66.98 | 61.03 | 71.04 |
| | DAL (2021b) | R-101 | 88.61 | 79.69 | 46.27 | 70.37 | 65.89 | 76.10 | 78.53 | 90.84 | 79.98 | 78.41 | 58.71 | 62.02 | 69.23 | 71.32 | 60.65 | 71.78 |
| | DRN (2020) | H-104 | 89.71 | 82.34 | 47.22 | 64.10 | 76.22 | 74.43 | 85.84 | 90.57 | 86.18 | 84.89 | 57.65 | 61.93 | 69.30 | 69.63 | 58.48 | 73.23 |
| | DCL (2021a) | R-152 | 89.10 | 84.13 | 50.15 | 73.57 | 71.48 | 58.13 | 78.00 | 90.89 | 86.64 | 86.78 | 67.97 | 67.25 | 65.63 | 74.06 | 67.05 | 74.06 |
| | GWD (2021c) | R-152 | 86.96 | 83.88 | 54.36 | 77.53 | 74.41 | 68.48 | 80.34 | 86.62 | 83.41 | 85.55 | 73.47 | 67.77 | 72.57 | 75.76 | 73.40 | 76.30 |
| | KFIoU (Ours) | R-152 | 89.46 | 85.72 | 54.94 | 80.37 | 77.16 | 69.23 | 80.90 | 90.79 | 87.79 | 86.13 | 73.32 | 68.11 | 75.23 | 71.61 | 69.49 | 77.35 |
| **Refine-stage** | R³Det (2021b) | R-152 | 89.80 | 83.77 | 48.11 | 66.77 | 78.76 | 83.27 | 87.84 | 90.82 | 85.38 | 85.51 | 65.67 | 62.68 | 67.53 | 78.56 | 72.62 | 76.47 |
| | CFA (2021) | R-152 | 89.08 | 83.20 | 54.37 | 66.87 | 81.23 | 80.96 | 87.17 | 90.21 | 84.32 | 86.09 | 52.34 | 69.94 | 75.52 | 80.76 | 67.96 | 76.67 |
| | RIDet (2021a) | R-50 | 89.31 | 80.77 | 54.07 | 76.38 | 79.81 | 81.99 | 89.13 | 90.72 | 83.58 | 87.22 | 64.42 | 67.56 | 78.08 | 79.17 | 62.07 | 77.62 |
| | S²A-Net (2021a) | R-50 | 88.89 | 83.60 | 57.74 | 81.95 | 79.94 | 83.19 | 89.11 | 90.78 | 84.87 | 87.81 | 70.30 | 68.25 | 78.30 | 77.01 | 69.58 | 79.42 |
| | R³Det-GWD (2021c) | R-152 | 89.66 | 84.99 | 59.26 | 82.19 | 78.97 | 84.83 | 87.70 | 90.21 | 86.54 | 86.85 | 73.47 | 67.77 | 76.92 | 79.22 | 74.92 | 80.23 |
| | R³Det-KLD (2021d) | R-152 | 89.92 | 85.13 | 59.19 | 81.33 | 78.82 | 84.38 | 87.50 | 89.80 | 87.33 | 87.00 | 72.57 | 71.35 | 77.12 | 79.34 | 78.68 | 80.63 |
| | R³Det-KFIoU (Ours) | Swin-T | 89.50 | 84.26 | 59.90 | 81.06 | 81.74 | 85.45 | 88.77 | 90.85 | 87.03 | 87.79 | 70.68 | 74.31 | 78.17 | 81.67 | 72.37 | 80.90 |
| | R³Det-KFIoU (Ours) | R-152 | 88.89 | 85.14 | 60.05 | 81.13 | 81.78 | 85.71 | 88.27 | 90.87 | 87.12 | 87.91 | 69.77 | 73.70 | 79.25 | 81.31 | 74.56 | 81.03 |
| **Two-stage** | RoI-Trans. (2019) | R-101 | 88.64 | 78.52 | 43.44 | 75.92 | 68.81 | 73.68 | 83.59 | 90.74 | 77.27 | 81.46 | 58.39 | 53.54 | 62.83 | 58.93 | 47.67 | 69.56 |
| | SCRDet (2019) | R-101 | 89.98 | 80.65 | 52.09 | 68.36 | 68.36 | 60.32 | 72.41 | 90.85 | 87.94 | 86.86 | 65.02 | 66.68 | 66.25 | 68.24 | 65.21 | 72.61 |
| | Gliding Vertex (2020) | R-101 | 89.64 | 85.00 | 52.26 | 77.34 | 73.01 | 73.14 | 86.82 | 90.74 | 79.02 | 86.81 | 59.55 | 70.91 | 72.94 | 70.86 | 57.32 | 75.02 |
| | CSL (2020) | R-152 | 90.25 | 85.53 | 54.64 | 75.31 | 70.44 | 73.51 | 77.62 | 90.84 | 86.15 | 86.69 | 69.60 | 68.04 | 73.83 | 71.10 | 68.93 | 76.17 |
| | RSDet-II (2021a) | R-152 | 89.93 | 84.45 | 53.77 | 74.35 | 71.52 | 78.31 | 78.12 | 91.14 | 87.35 | 86.93 | 65.64 | 65.17 | 75.35 | 79.74 | 63.31 | 76.34 |
| | SCRDet++ (2023) | R-101 | 90.05 | 84.39 | 55.44 | 73.99 | 77.54 | 71.11 | 86.05 | 90.67 | 87.32 | 87.08 | 69.62 | 68.90 | 73.74 | 71.29 | 65.08 | 76.81 |
| | ReDet (2021b) | ReR-50 | 88.81 | 82.48 | 60.83 | 80.82 | 78.34 | 86.06 | 88.31 | 90.87 | 88.77 | 87.03 | 68.65 | 66.90 | 79.26 | 79.71 | 74.67 | 80.10 |
| | Oriented R-CNN (2021) | R-50 | 89.84 | 85.43 | 61.09 | 79.82 | 79.71 | 85.35 | 88.82 | 90.88 | 86.68 | 87.73 | 72.21 | 70.80 | 82.42 | 78.18 | 74.11 | 80.87 |
| | RoI-Trans.-KFIoU (Ours) | Swin-T | 89.44 | 84.41 | 62.22 | 82.51 | 80.10 | 86.07 | 88.68 | 90.90 | 87.32 | 88.38 | 72.80 | 71.95 | 78.96 | 74.95 | 75.27 | 80.93 |

the model show strong performance in large aspect ratio objects (e.g. SV, LV, SH). However, the large number of anchors makes it time-consuming. Therefore, we use horizontal anchors by default to balance accuracy and speed. In terms of definition, experiments show that OpenCV definition ($D_{oc}$) (Yang et al., 2019) is slightly better than Long Edge definition ($D_{le}$) (Ma et al., 2018) on the three versions of DOTA. Angle direct regression (Reg.) always suffers from the boundary discontinuity problem as widely studied recently (Yang & Yan, 2020). In contrast, angle indirect regression (Reg*.) is a simpler way to avoid above issues and brings performance boost according to Tab. 4.

PIoU calculates the SkewIoU by accumulating the contribution of interior overlapping pixels but the effect is not significant. IoU-Smooth L1 partly circumvents the need for SkewIoU loss with gradient backpropagation by combining IoU and Smooth L1 loss. Although IoU-Smooth L1 has achieved an improvement of 1.26%/0.29%/2.15% on DOTA-v1.0/v1.5/v2.0, the gradient is still dominated by Smooth L1 but still worse than plain SkewIoU loss. Modulated Loss and RIL implement ordered and disordered quadrilateral detection respectively, and the more accurate representation makes them both have a considerable performance improvement. In particular, Modulated Loss achieves the third highest performance on DOTA-v1.5/v2.0. CSL and DCL convert the angle prediction from regression to classification, cleverly eliminating the boundary discontinuity problem caused by the angle periodicity. GWD loss, KLD loss and KFIoU loss are three different regression losses based on Gaussian distribution. The results presented in our experiments of GWD loss and KLD loss are obtained by best-tuned hyperparameters. In contrast, KFIoU loss is free from hyperparameter tuning and has a more stable performance increase due to a more consistent calculation process with SkewIoU loss as the center point gets closer.

## 5.3 COMPARISON WITH THE STATE-OF-THE-ART

Tab. 5 compares recent detectors on DOTA-v1.0, as categorized by single-, refine-, and two-stage based methods. Since different methods use different image resolution, network structure, training strategies and various tricks, we cannot make absolutely fair comparisons. In terms of overall performance, our method has achieved the best performance so far, at around **77.35%/81.03%/80.93%**.

## 6 CONCLUSION

We have presented a trend-level consistent approximate to the ideal but gradient-training unfriendly SkewIoU loss for rotation detection, and we call it KFIoU loss as the product of the Gaussian distributions is adopted to directly mimic the computing mechanism of SkewIoU loss by definition. This design differs from the distribution distance based losses including GWD loss and KLD loss which in our analysis have fundamental difficulty in achieving trend-level alignment with SkewIoU loss without tuning hyperparameters. Moreover, KFIoU is easier to implement and works better than plain SkewIoU due to the effective measurement for non-overlapping cases and complete derivation. Experimental results on various 2D and 3D datasets show the effectiveness of our approach.

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

## A  PROOF OF KFIoU UPPER BOUND

For an n-dimensional Gaussian distribution, its volume is:

$$\mathcal{V} = 2^n \cdot |\mathbf{\Sigma}^{\frac{1}{2}}| = 2^n \cdot |\mathbf{\Sigma}|^{\frac{1}{2}} \tag{14}$$

For $\mathbf{\Sigma}_{kf}$, we have

$$
\begin{aligned}
|\mathbf{\Sigma}_{kf}| =& |\mathbf{\Sigma}_1 - \mathbf{\Sigma}_1(\mathbf{\Sigma}_1 + \mathbf{\Sigma}_2)^{-1}\mathbf{\Sigma}_1| \\
=& |\mathbf{\Sigma}_1(\mathbf{\Sigma}_1 + \mathbf{\Sigma}_2)^{-1}\mathbf{\Sigma}_2| \\
=& \frac{|\mathbf{\Sigma}_1| \cdot |\mathbf{\Sigma}_1|}{|\mathbf{\Sigma}_1 + \mathbf{\Sigma}_2|}
\end{aligned}
\tag{15}
$$

According to Minkowski's inequality:

$$|\mathbf{\Sigma}_1 + \mathbf{\Sigma}_2|^{\frac{1}{n}} \geq |\mathbf{\Sigma}_1|^{\frac{1}{n}} + |\mathbf{\Sigma}_2|^{\frac{1}{n}} \tag{16}$$

Simultaneous mean inequalities:

$$|\mathbf{\Sigma}_1 + \mathbf{\Sigma}_2|^{\frac{1}{n}} \geq |\mathbf{\Sigma}_1|^{\frac{1}{n}}| + |\mathbf{\Sigma}_2|^{\frac{1}{n}} \geq 2 \cdot |\mathbf{\Sigma}_1|^{\frac{1}{2n}} \cdot |\mathbf{\Sigma}_2|^{\frac{1}{2n}} \tag{17}$$

Thus:

$$
\begin{aligned}
\frac{|\mathbf{\Sigma}_1|^{\frac{1}{2n}} \cdot |\mathbf{\Sigma}_2|^{\frac{1}{2n}}}{|\mathbf{\Sigma}_1 + \mathbf{\Sigma}_2|^{\frac{1}{n}}} \leq& \frac{1}{2} \\
\frac{|\mathbf{\Sigma}_1|^{\frac{1}{2}} \cdot |\mathbf{\Sigma}_2|^{\frac{1}{2}}}{|\mathbf{\Sigma}_1 + \mathbf{\Sigma}_2|} \leq& \frac{1}{2^n}
\end{aligned}
\tag{18}
$$

and

$$
\begin{aligned}
|\mathbf{\Sigma}_{kf}| =& \frac{|\mathbf{\Sigma}_1| \cdot |\mathbf{\Sigma}_1|}{|\mathbf{\Sigma}_1 + \mathbf{\Sigma}_2|} \leq \frac{|\mathbf{\Sigma}_1|^{\frac{1}{2}} \cdot |\mathbf{\Sigma}_2|^{\frac{1}{2}}}{2^n} \\
|\mathbf{\Sigma}_{kf}|^{\frac{1}{2}} \leq& \frac{|\mathbf{\Sigma}_1|^{\frac{1}{4}} \cdot |\mathbf{\Sigma}_2|^{\frac{1}{4}}}{2^{\frac{n}{2}}}
\end{aligned}
\tag{19}
$$

Combine the mean inequalities again:

$$|\mathbf{\Sigma}_{kf}|^{\frac{1}{2}} \leq \frac{|\mathbf{\Sigma}_1|^{\frac{1}{4}} \cdot |\mathbf{\Sigma}_2|^{\frac{1}{4}}}{2^{\frac{n}{2}}} \leq \frac{|\mathbf{\Sigma}_1|^{\frac{1}{2}} + |\mathbf{\Sigma}_2|^{\frac{1}{2}}}{2^{\frac{n}{2}+1}} \tag{20}$$

According to Eq. 14, we have

$$\mathcal{V}_{kf} \leq \frac{\mathcal{V}_1 + \mathcal{V}_2}{2^{\frac{n}{2}+1}} \tag{21}$$

Therefore, the upper bound of KFIoU is

$$\text{KFIoU} = \frac{\mathcal{V}_{kf}}{\mathcal{V}_1 + \mathcal{V}_2 - \mathcal{V}_{kf}} \leq \frac{1}{2^{\frac{n}{2}+1} - 1} \tag{22}$$

When $n = 2$ and $n = 3$, the upper bounds are $\frac{1}{3}$ and $\frac{1}{\sqrt{32}-1}$ respectively.

## B  SUPPLEMENTARY EXPERIMENT

**Ablation study of three forms of KFIoU loss on two detectors.** We use two different detectors and three different KFIoU based loss functions to verify its effectiveness, as shown in Tab. 6. RetinaNet-based detector will have a large number of low-SkewIoU prediction bounding box in the early stage of training, and will produce very large loss after the log function, which weakens the improvement of the model. Compared with the linear function, the derivative of the exp-based function will pay more attention to the training of difficult samples, so it has a higher performance, at **70.64%**. In contrast, R³Det-based detector can generate high-quality prediction box at the beginning of training by adding refinement stages, so it will not suffer the same troubles as RetinaNet. Due to the same mechanism of focusing on difficult samples, log and exp-based functions are both better than linear functions, and the best performance is achieved on the log-based function, about **72.28%**. We also expand KFIoU by 3 times to make its range truly consistent with the IoU loss, at [0, 1]. However, this

Table 6: Ablation study of different KFIoU loss forms with different detectors on DOTA-v1.0.

| Method | Smooth L1 | $-\ln(\mathbf{KFIoU}+\epsilon)$ | $1-\mathbf{KFIoU}$ | $e^{1-\mathbf{KFIoU}}-1$ | $e^{1-3\mathbf{KFIoU}}-1$ | $-\ln(3\mathbf{KFIoU}+\epsilon)$ |
|---|---|---|---|---|---|---|
| RetinaNet | 65.73 | 69.80 (+4.07) | 70.19 (+4.46) | **70.64 (+4.91)** | 69.64 (+3.91) | – |
| R$^3$Det | 70.66 | **72.28 (+1.62)** | 71.09 (+0.43) | 71.58 (+0.92) | – | 71.77 (+1.11) |

Table 7: Ablation study of training strategies and tricks. Rotate and MS indicate rotation augmentation and multi-scale training and testing.

| Method | KFIoU | Backbone | Sched. | MS | Rotate | PL | BD | BR | GTF | SV | LV | SH | TC | BC | ST | SBF | RA | HA | SP | HC | mAP$_{50}$ |
|---|---|---|---|---|---|---|---|---|---|---|---|---|---|---|---|---|---|---|---|---|---|
| RetinaNet |  | R-50 | 12e |  |  | 87.76 | 72.61 | 43.86 | 66.61 | 69.70 | 56.61 | 74.15 | 90.86 | 75.27 | 79.09 | 47.81 | 64.60 | 58.93 | 63.37 | 26.58 | 65.19 |
|  | ✓ | R-50 | 12e |  |  | 88.90 | 80.68 | 47.12 | 70.40 | 72.20 | 62.49 | 74.84 | 90.91 | 79.63 | 79.73 | 58.54 | 66.40 | 63.67 | 67.13 | 45.33 | 69.86 |
| S$^2$A-Net |  | R-50 | 12e |  |  | 89.18 | 79.35 | 49.11 | 72.97 | 79.08 | 78.03 | 86.67 | 90.91 | 85.90 | 85.04 | 64.06 | 65.64 | 66.71 | 67.60 | 48.08 | 73.89 |
|  | ✓ | R-50 | 12e |  |  | 89.24 | 83.46 | 51.44 | 70.88 | 78.70 | 76.31 | 86.90 | 90.90 | 82.22 | 84.81 | 61.67 | 66.93 | 65.62 | 67.99 | 57.05 | 74.27 |
| RoI Trans. |  | R-50 | 12e |  |  | 89.02 | 81.71 | 53.84 | 71.65 | 79.00 | 77.76 | 87.85 | 90.90 | 87.04 | 85.70 | 61.73 | 64.55 | 75.06 | 71.71 | 62.38 | 75.99 |
|  | ✓ | R-50 | 12e |  |  | 89.08 | 82.62 | 53.90 | 71.78 | 78.73 | 77.91 | 87.97 | 90.90 | 86.68 | 85.37 | 63.17 | 67.65 | 74.30 | 71.19 | 61.35 | 76.17 |
|  |  | Swin-T | 12e |  |  | 88.96 | 82.81 | 53.34 | 76.55 | 78.66 | 83.54 | 88.00 | 90.90 | 86.95 | 86.47 | 41.94 | 64.17 | 76.29 | 72.87 | 63.95 | 77.18 |
|  | ✓ | Swin-T | 12e |  |  | 88.9 | 83.77 | 53.98 | 77.63 | 78.83 | 84.22 | 88.15 | 90.91 | 87.21 | 86.14 | 67.79 | 65.73 | 75.80 | 73.68 | 63.30 | 77.74 |
|  | ✓ | R-50 | 24e | ✓ | ✓ | 89.12 | 84.54 | 60.73 | 78.86 | 79.65 | 85.79 | 88.45 | 90.90 | 87.03 | 88.28 | 69.15 | 70.28 | 78.88 | 81.54 | 70.05 | 80.22 |
|  | ✓ | Swin-T | 12e | ✓ | ✓ | 89.44 | 84.41 | 62.22 | 82.51 | 80.10 | 86.07 | 88.68 | 90.90 | 87.32 | 88.38 | 72.80 | 71.95 | 78.96 | 74.95 | 75.27 | **80.93** |
| R$^3$Det | ✓ | R-50 | 12e |  |  | 89.02 | 74.52 | 47.93 | 69.64 | 77.02 | 74.07 | 82.56 | 90.90 | 79.39 | 83.67 | 59.02 | 62.51 | 63.56 | 65.06 | 37.22 | 70.41 |
|  | ✓ | R-50 | 12e |  |  | 89.06 | 73.89 | 49.82 | 68.39 | 78.13 | 75.35 | 86.65 | 90.89 | 82.57 | 83.84 | 59.63 | 62.03 | 66.16 | 66.22 | 47.98 | 72.04 |
|  | ✓ | R-50 | 12e | ✓ |  | 89.06 | 82.49 | 55.91 | 81.04 | 80.14 | 83.24 | 88.56 | 90.90 | 84.61 | 86.83 | 66.25 | 71.50 | 75.60 | 77.64 | 63.66 | 78.50 |
|  | ✓ | Swin-T | 12e | ✓ |  | 89.41 | 83.66 | 56.92 | 79.76 | 80.45 | 84.34 | 88.71 | 90.91 | 85.69 | 87.64 | 67.69 | 72.88 | 76.34 | 73.63 | 72.21 | 79.35 |
|  | ✓ | R-50 | 12e | ✓ | ✓ | 89.33 | 84.19 | 58.78 | 81.30 | 80.48 | 84.49 | 88.85 | 90.84 | 85.56 | 87.57 | 69.14 | 70.79 | 77.33 | 80.82 | 66.51 | 79.73 |
|  | ✓ | R-101 | 12e | ✓ | ✓ | 89.28 | 83.32 | 59.40 | 80.29 | 80.43 | 84.70 | 88.85 | 90.87 | 84.51 | 87.95 | 71.86 | 71.60 | 78.31 | 79.42 | 66.60 | 79.83 |
|  | ✓ | Swin-T | 12e | ✓ | ✓ | 89.24 | 83.75 | 59.77 | 79.40 | 80.95 | 84.61 | 88.84 | 90.84 | 86.86 | 87.93 | 71.71 | 71.17 | 76.79 | 77.42 | 71.59 | 80.06 |
|  | ✓ | Swin-T | 24e | ✓ | ✓ | 89.50 | 84.26 | 59.90 | 81.06 | 81.74 | 85.45 | 88.77 | 90.85 | 87.03 | 87.79 | 70.68 | 74.31 | 78.17 | 81.67 | 72.37 | **80.90** |

Table 8: Results on KITTI val split 3D detection and BEV Detection.

| Method | mAP | Car - 3D Detection | | | Ped. - 3D Detection | | | Cyc. - 3D Detection | | |
|---|---|---|---|---|---|---|---|---|---|---|
|  | Mod. | Easy | Mod. | Hard | Easy | Mod. | Hard | Easy | Mod. | Hard |
| PointPillars | 64.28 | 88.26 | 78.90 | 76.06 | 57.10 | 50.96 | 46.38 | 83.77 | 62.99 | 59.65 |
| +GWD | 65.50 | 87.38 | 78.57 | 75.87 | 61.69 | 55.19 | 50.04 | 81.61 | 62.74 | 59.18 |
| +KLD | 66.19 | 89.55 | 80.36 | 76.02 | 59.95 | 52.94 | 48.22 | 85.61 | 65.27 | 61.45 |
| +KFIoU | **66.71** | 89.56 | 80.19 | 77.16 | 60.97 | 54.94 | 50.75 | 84.96 | 65.00 | 61.00 |

| Method | mAP | Car - BEV Detection | | | Ped. - BEV Detection | | | Cyc. - BEV Detection | | |
|---|---|---|---|---|---|---|---|---|---|---|
|  | Mod. | Easy | Mod. | Hard | Easy | Mod. | Hard | Easy | Mod. | Hard |
| PointPillars | 70.10 | 93.81 | 88.08 | 86.80 | 61.49 | 55.51 | 51.13 | 87.20 | 66.69 | 63.02 |
| +GWD | 71.48 | 92.02 | 88.30 | 85.72 | 64.67 | 58.49 | 53.45 | 86.92 | 67.66 | 63.37 |
| +KLD | 71.18 | 93.33 | 88.11 | 85.44 | 64.46 | 57.26 | 52.53 | 87.40 | 68.19 | 64.47 |
| +KFIoU | **72.08** | 92.15 | 89.90 | 85.66 | 63.45 | 57.81 | 53.07 | 87.52 | 68.55 | 64.56 |

consistency do not bring any additional gains, so the following experiments are all use the KFIoU before non-expansion.

**Ablation study of training strategies and tricks.** We reimplement KFIoU based on the more powerful benchmark, MMRotate (Zhou et al., 2022). We use a single GeForce RTX 3090 Ti with a total batch size of 2 for training. For ResNet (He et al., 2016), SGD optimizer is adopted with an initial learning rate of 0.0025. The momentum and weight decay are 0.9 and 0.0001, respectively. For Swin Transformer (Liu et al., 2021), AdamW (Kingma & Ba, 2014; Loshchilov & Hutter, 2018) optimizer is adopted with an initial learning rate of 0.0001. The weight decay is 0.05. In addition, we adopt learning rate warmup for 500 iterations, and the learning rate is divided by 10 at each decay step. Tab. 7 performs ablation experiments on four detectors: RetinaNet (Lin et al., 2017b), S$^2$A-Net (Han et al., 2021a), R$^3$Det (Yang et al., 2021b), and RoI Transformer (Ding et al., 2019). The experimental results prove that KFIoU can stably enhance the performance of the detector. In order to further improve the performance of the model on DOTA, we verified many commonly used training strategies and tricks, including backbone, training schedule, data augmentation and multi-scale training and testing, as shown in Tab. 7.

**Ablation study of KFIoU loss on 3-D case.** More detailed results in KITTI are shown in Tab. 8.

**Ablation study on more datasets.** The performance of different loss functions is compared in Tab. 9 on ICDAR2015, UCAS-AOD, SSDD (Li et al., 2017) and HRSID (Wei et al., 2020b) datasets, and KFIoU is still the best.

Table 9: Results on more datasets, the base detector is RetinaNet.

| Loss | ICDAR2015 | UCAS-AOD | | | SSDD | HRSID |
|---|---|---|---|---|---|---|
| | | Car | Plane | mAP$_{50}$ | Inshore | Inshore |
| Smooth L1 | 69.78 | 92.62 | 96.50 | 94.56 | 68.47 | 51.41 |
| GWD | 74.29 | 94.03 | 96.86 | 95.44 | 77.71 | 51.11 |
| KLD | 75.32 | 94.34 | 97.94 | 96.14 | 76.84 | 52.80 |
| KFIoU | **75.90** | **94.51** | **98.41** | **96.46** | **77.89** | **53.45** |

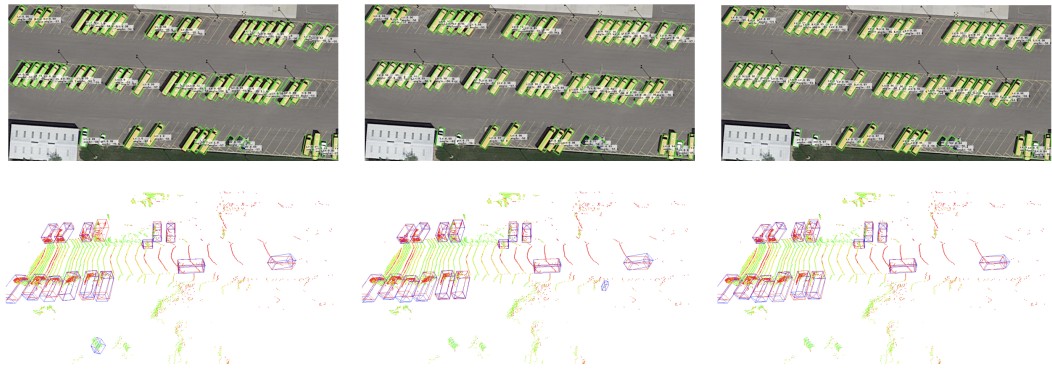

Figure 4: Visual comparison between Smooth L1 loss-based (**left**), GWD-based (**middle**) and the KFIoU-based (**right**) detectors on DOTA (2-D) and KITTI (3-D). For 3-D object detection, red and blue box denotes ground-truth and predict bounding box, respectively.

## C  VISUALIZATION

Fig. 4 ad Fig. 5 show the visual comparison of three different loss functions on the different kinds of datasets. Compared with Smooth L1 Loss, KFIoU loss is significantly better.

## D  TREND CONSISTENCY SIMULATION

Fig. 6(a) and Fig. 6(b) show the impact of center deviation and object scale on the trend consistency of each loss function. Note that each data in the figure is calculated from the average of 1,000 random aspect ratio and rotation angle examples. Two conclusions can be drawn: i) the smaller the center deviation, the better trend consistency of the KFIoU loss; ii) KLD loss and KFIoU loss are insensitive to scale changes.

## E  LIMITATION

Note that the Gaussian modeling has a limitation that it cannot be directly applied to quadrilateral/polygon detection (Ming et al., 2021a; Xu et al., 2020) which is also an important task in aerial images, scene text, etc. In addition, the Gaussian distribution of the square like object is close to the isotropic circle, which is not suitable for the object heading detection.

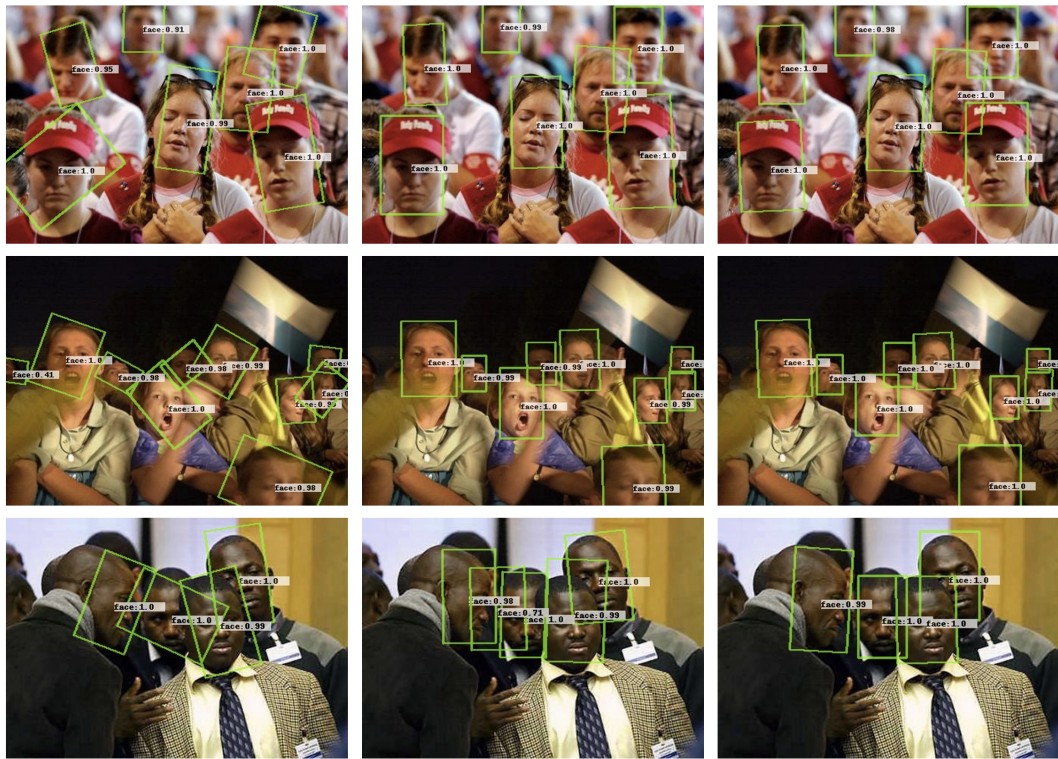

Figure 5: Visual comparison between Smooth L1 loss-based (**left**), GWD-based (**middle**) and the KFIoU-based (**right**) detectors on FDDB.

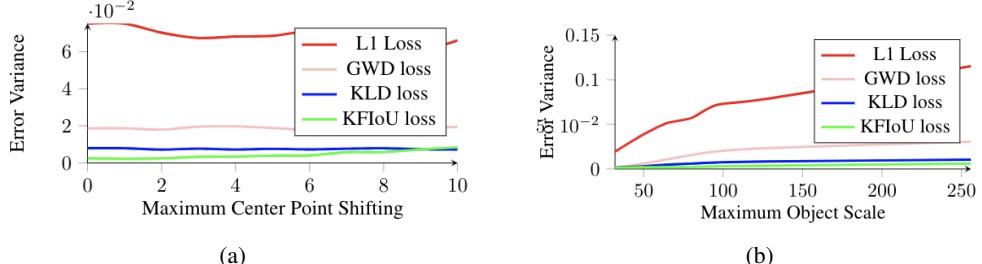

(a)                                                             (b)

Figure 6: (a) Impact of center deviation on the trend consistency of each loss function. (b) Impact of object scale on the trend consistency of each loss function under a 5 pixels center deviations.

