# OpenReview forum: "The KFIoU Loss for Rotated Object Detection"
_ICLR.cc/2023/Conference — ICLR 2023 poster_

### Official Review · Reviewer_jv8Z · 2022-10-24

**Confidence:** 4
**Correctness:** 4
**Technical Novelty And Significance:** 3
**Empirical Novelty And Significance:** 3
**Recommendation:** 8

**Clarity, Quality, Novelty And Reproducibility:**

Although many building blocks of the proposed approach already exist in previous works.
The paper is well-motivated and the writing is clear. Ablation study and comparison are comprehensive.


**Strength And Weaknesses:**

STRENGTH
- The adoptation of Kalman filter formulation to compute the overlapping in rotated objects is interesting.
- The contributions and roles of each building block are analyzed and discussed.
- Ablation study and comparison are comprehensive.
- Experiments on both 2D/3D data have shown the advantages of the proposed approach.

WEAKNESS
In Table 2, only L1 smooth is used as baseline.
While the proposed approach give comparable/marginal results with KLD, more comparison with KLD in ablation study is recommended.

**Summary Of The Paper:**

In this paper, the authors introduce a KFIoU loss for rotated object detection.
The general idea of KFIoU loss is to approximate the SkewIoU based on Gaussian Modeling and Kalman filter.
Particularly, a bounding box is firstly represented as a Gaussian distribution form.
Then (1) a scale-insensitive center point loss to measure the distance between the centers of Gaussians and (2) a Kalman-based term to measure the overlapping between Gaussian distributions are adopted.
Finally, these measurements are converted back to bounding box form for approximating the SkewIoU.
The proposed approach is validated on several detection benchmarks in both 2D and 3D cases to illustrate its advantages.

**Summary Of The Review:**

Although many building blocks of the proposed approach already exist in previous works.
All modifications and contributions have been discusses.
Ablation study and comparison are comprehensive.
Experiments have shown the advantages of the proposed approaches.

---

> ### Author Response · Authors · 2022-11-10
> **Response to Reviewer jv8Z (Round 1)**
>
> > ***Q1: In Table 2, only L1 smooth is used as baseline. While the proposed approach give comparable/marginal results with KLD, more comparison with KLD in ablation study is recommended.***
>
> **A1:** Thank you for your valuable suggestion. We will try our best to supplement more experiments compared with KLD. The following is a part of the current supplement (staying producing more results during the rebuttal):
>
> | 2D Dataset | Loss      | Hmean     |
> | ---------- | --------- | --------- |
> | ICDAR2015  | Smooth L1 | 69.78     |
> | ICDAR2015  | GWD       | 74.29     |
> | ICDAR2015  | KLD       | 75.32     |
> | ICDAR2015  | KFIoU     | **75.90** |
>
> | 2D Dataset | Loss      | Car       | Plane     | mAP       |
> | ---------- | --------- | --------- | --------- | --------- |
> | UCAS-AOD   | Smooth L1 | 92.62     | 96.50     | 94.56     |
> | UCAS-AOD   | GWD       | 94.03     | 96.86     | 95.44     |
> | UCAS-AOD   | KLD       | 94.34     | 97.94     | 96.14     |
> | UCAS-AOD   | KFIoU     | **94.51** | **98.41** | **96.46** |
>
> | Loss      | SSDD inshore | HRSID inshore |
> | --------- | ------------ | ------------- |
> | Smooth L1 | 68.47        | 51.41         |
> | GWD       | 77.71        | 51.11         |
> | KLD       | 76.84        | 52.80         |
> | KFIoU     | **77.89**    | **53.45**     |
>
> | 3D Dataset | Method       | 3-D Mod. mAP | BEV Mod. mAP |
> | ---------- | ------------ | ------------ | ------------ |
> | KITTI      | PointPillars | 64.28        | 70.10        |
> | KITTI      | +GWD         | 65.50        | 71.48        |
> | KITTI      | +KLD         | 66.19        | 71.18        |
> | KITTI      | +KFIoU       | **66.71**    | **72.08**    |

---

### Official Review · Reviewer_Q8Fo · 2022-11-03

**Confidence:** 2
**Correctness:** 3
**Technical Novelty And Significance:** 3
**Empirical Novelty And Significance:** 3
**Recommendation:** 6

**Clarity, Quality, Novelty And Reproducibility:**


The authors proposed a novel approach, focusing on the loss function. The use of Gaussian-modeling-based losses to approximate the SkewIoU loss is not new, but the use of Kalman filters for this use is, and seems to be relevant.

One of the major arguments of the authors is that their proposed loss does not involve any additional hyperparameter, contrary to the GWD and KLW losses. It remains unclear how this is a drawback. What is the concrete effort to tune these hyperparameters ? Does it exist default values for them, and what is the impact on the final performance (best-tuned vs default) ? In brief, are these hyperparameters a real issue ?

Straight arrows from Figure 1 are hard to understand. Why not plotting the smooth l1 loss directly on the graph ?

The authors will make the code public for reproducibility.

**Strength And Weaknesses:**

Strength:

Contrary to other SkewIoU loss approximations in the literature (GWD/KLD), the proposed KFIoU loss does not involve any new hyperparameter.

The proposed KFIoU loss enabled the authors to reach competitive/SOTA results on the DOTA dataset. The ablation study show that the FKIoU loss achieves better results than the smooth L1 loss on three detection tasks (aerial, scene text, and face).
The results on the 3D detection seem also promising.

Weaknesses:

Some results do not seem fair compared to SOTA.
Table 1 shows that GWD and KLD reach better results than the smooth L1 loss on the DOTA datasets. However, in the ablation study of table 2, where the approach is evaluated on the other 2D datasets, the proposed loss is only compared with the smooth L1 loss. Why not comparing with GWD and KLD ?
In Table 5, when comparing the 2-stage approaches, the backbone used is different from all the other works, preventing the fair comparison. One can ask if the improvement is due to the loss, or to the backbone.

Be careful with the proofreading, some errors complicate the reading (e.g. logical connectors in the beginning of the abstract).

The limitation mentioned in the appendix should be highlighted in a dedicated section, before the conclusion, for example.


**Summary Of The Paper:**

The authors proposed a new loss, called KFIoU, for the detection of rotated objects. Based on Kalman filters and Gaussian modeling, this loss aims at approximating the SkewIoU loss, which is unfriendly to differentiable learning. Formulated this way, the SkewIoU loss does not rely on any additional hyperparameter, contrary to its counterparts. Moreover, experiments show that the proposed loss is nearer to the SkewIoU loss than the other approximator counterparts of the literature in terms of error variance.

**Summary Of The Review:**

The KFIoU loss proposed by the authors seems interesting compared to other SkewIoU estimators, while not relying on additional hyperparameters. While the approach reaches SOTA results on the DOTA dataset, some choices are questionable regarding the other comparisons, making it difficult to compare with the state of the art on the other 2D datasets.

---

> ### Author Response · Authors · 2022-11-10
> **Response to Reviewer Q8Fo (Round 1)**
>
> > ***Q1: Some results do not seem fair compared to SOTA. Table 1 shows that GWD and KLD reach better results than the smooth L1 loss on the DOTA datasets. However, in the ablation study of table 2, where the approach is evaluated on the other 2D datasets, the proposed loss is only compared with the smooth L1 loss. Why not comparing with GWD and KLD?***
>
> **A1:** Thank you for your insightful suggestion. We will try our best to supplement more experiments compared with KLD. The following is a part of our current new results (staying producing more results):
>
> | 2D Dataset | Loss      | Hmean     |
> | ---------- | --------- | --------- |
> | ICDAR2015  | Smooth L1 | 69.78     |
> | ICDAR2015  | GWD       | 74.29     |
> | ICDAR2015  | KLD       | 75.32     |
> | ICDAR2015  | KFIoU     | **75.90** |
>
> | 2D Dataset | Loss      | Car       | Plane     | mAP       |
> | ---------- | --------- | --------- | --------- | --------- |
> | UCAS-AOD   | Smooth L1 | 92.62     | 96.50     | 94.56     |
> | UCAS-AOD   | GWD       | 94.03     | 96.86     | 95.44     |
> | UCAS-AOD   | KLD       | 94.34     | 97.94     | 96.14     |
> | UCAS-AOD   | KFIoU     | **94.51** | **98.41** | **96.46** |
>
> | Loss      | SSDD inshore | HRSID inshore |
> | --------- | ------------ | ------------- |
> | Smooth L1 | 68.47        | 51.41         |
> | GWD       | 77.71        | 51.11         |
> | KLD       | 76.84        | 52.80         |
> | KFIoU     | **77.89**    | **53.45**     |
>
> | 3D Dataset | Method       | 3-D Mod. mAP | BEV Mod. mAP |
> | ---------- | ------------ | ------------ | ------------ |
> | KITTI      | PointPillars | 64.28        | 70.10        |
> | KITTI      | +GWD         | 65.50        | 71.48        |
> | KITTI      | +KLD         | 66.19        | 71.18        |
> | KITTI      | +KFIoU       | **66.71**    | **72.08**    |
>
> > ***Q2: In Table 5, when comparing the 2-stage approaches, the backbone used is different from all the other works, preventing the fair comparison. One can ask if the improvement is due to the loss, or to the backbone.***
>
> **A2:** The structure of different methods and the tricks used in Table 5 are different, so it is difficult to ensure true fairness. Indeed, KFIoU loss cannot alone make RoI Trans. achieve the best performance, because RoI Trans. is not the most SOTA detector. Therefore, we use a stronger backbone. In addition, please note that Table 7 in the appendix contains relevant ablation experiments, that is, KFIoU loss can improve the performance of RoI Trans.
>
> > ***Q3: The limitation mentioned in the appendix should be highlighted in a dedicated section, before the conclusion, for example.***
>
> **A3:** We have changed the "Conclusion" section into "Discussion" section (including limitation and conclusion) in the paper.
>
> > ***Q4: What is the concrete effort to tune these hyperparameters? Does it exist default values for them, and what is the impact on the final performance (best-tuned vs default)? In brief, are these hyperparameters a real issue?***
>
> **A4:** GWD and KLD need extra hyperparameters tuning and metric selection that vary across datasets and detectors, thus it is difficult to find a default value to achieve best performance, and it takes a lot of time to tune. Compared to best-tuned GWD or KLD, our proposed method achieves more competitive performance without cost much to tune hyperparameter. Therefore, hyperparameters tuning is a real issue.
>
> > ***Q5: Straight arrows from Figure 1 are hard to understand. Why not plotting the smooth l1 loss directly on the graph?***
>
> **A5:** We have improved the readibility of Fig. 1 per your valuable suggestion. Please refer to Figure 3(a) and Figure 3(b) for the smooth l1 loss.

---

> > ### Comment · Reviewer_Q8Fo · 2022-11-25
> > **Response to authors**
> >
> > Thank you for your responses.
> > The additional results are consistent, and your clarifications are satisfactory.
> >
> > Based on the other reviewers' comments and your responses, I maintain my previous rating (6: marginally above the acceptance threshold).

---

> > > ### Author Response · Authors · 2022-11-25
> > > **Response to Reviewer Q8Fo**
> > >
> > > Thank you.

---

### Official Review · Reviewer_2M4a · 2022-11-03

**Confidence:** 3
**Correctness:** 3
**Technical Novelty And Significance:** 3
**Empirical Novelty And Significance:** 3
**Recommendation:** 6

**Clarity, Quality, Novelty And Reproducibility:**

The paper is well written for the most parts. There are a few typos (e.g. equation (6)’s second term is incorrect as is, and is probably unnecessary as well), and some structural changes might benefit the paper (e.g. wait until eq. (7) to discuss bounding box volume, write out explicitly the full loss in a dedicated equation).
As discussed in the weaknesses, I think the paper overall suffers from leaning so heavily on Kalman filtering. Even without making use of Kalman filtering, I feel like the contribution of the paper is significant enough.
Miscellaneous comments:
- Why not show the EMean in Table 1?
- Is the upper bound on the value of KFIoU really important? An estimation of the difference between SkewIoU and KFIoU – if attainable – would be much more valuable.


**Strength And Weaknesses:**

Strengths:
- KFIoU is an easy to compute and differentiate loss with well principled motivation.
- KFIoU is hyper-parameter free and fairly simple to implement.
- The paper is mostly well-written with clear illustrations (notably Fig. 2) that sum up the problem and solution.
- Experimental results seem to validate the efficiency of the method with regards to previous approaches on both 2D and 3D detection tasks.

Weaknesses:
- No indication of uncertainty or context is given on the results. What is the standard deviation? How many runs were performed for each result? Are the baseline results reported or reproduced? If reported, are the settings comparable?
- The center loss is simply added as a correction term. This seems to work well, but it does not really reflect the product of Gaussian at the center of the paper. Have the authors tried using the distance between centers as a multiplicative factor on the KFIoU?
- This is more of an issue of semantics, but I am unconvinced by the paper’s positioning around Kalman filtering. At its core, KFIoU computes the overlap between two Gaussians through a product of the pdfs which is well known to also be a Gaussian pdf up to a scale factor. While this procedure is indeed very important in Kalman filtering to reconcile observations and predictions, it is just a small part of the entire process. As such, I feel like the paper’s reliance on Kalman filtering detracts from its actual key points.

**Summary Of The Paper:**

Training deep models to correctly detect and outline rotated objects is difficult due to the nature of the metric of reference (SkewIoU). The authors propose KFIoU, an easily computable and differentiable loss that relies on transforming bounding boxes into Gaussians and estimating the overlap between the Gaussians analytically. Experiments demonstrate both that the derived proxy loss follows SkewIoU trends fairly well, and leads to strong performance on both 2D and 3D benchmarks.


**Summary Of The Review:**

KFIoU is an interesting and easy to implement proxy for SkewIoU that follows easy to understand principles. Experimental results show the loss outperforms much of its competition and even sets a new state of the art on the DOTA dataset. As such, I recommend acceptance for this paper.

---

> ### Author Response · Authors · 2022-11-10
> **Response to Reviewer 2M4a (Round 1, Part 2/2)**
>
> > ***Q3: This is more of an issue of semantics, but I am unconvinced by the paper’s positioning around Kalman filter. At its core, KFIoU computes the overlap between two Gaussians through a product of the pdfs which is well known to also be a Gaussian pdf up to a scale factor. While this procedure is indeed very important in Kalman filter to reconcile observations and predictions, it is just a small part of the entire process. As such, I feel like the paper’s reliance on Kalman filter detracts from its actual key points.***
>
> **A3:** We fully agree with you. The core of KFIoU loss is the product of the Gaussian distributions, rather than the whole process of Kalman filter. The reason why it is named KFIoU is that our inspiration comes from Kalman filter, so we chose this name to pay tribute to it. **In order to prevent misleading readers, we have revised the paper, replacing "Kalman filter" with the keyword "Gaussian product", and explained the reason for the name KFIoU loss in the footnote on page 2.**
>
> > ***Q4: There are a few typos (e.g. equation (6)’s second term is incorrect as is, and is probably unnecessary as well), and some structural changes might benefit the paper (e.g. wait until eq. (7) to discuss bounding box volume, write out explicitly the full loss in a dedicated equation).***
> >
> **A4:** Thank you for your reminder, we have modified the Eq. (6) as $\alpha=N_{\mu_{1}}(\mu_{2},\Sigma_{1}+\Sigma_{2})$, and Eq. (5) as $\alpha N_{x}(\mu,\Sigma)=N_{x}(\mu_{1},\Sigma_{1})N_{x}(\mu_{2},\Sigma_{2})$ according to Section 8.1.8 in http://www2.imm.dtu.dk/pubdb/edoc/imm3274.pdf.
>
> > ***Q5: Why not show the EMean in Table 1?***
> >
> **A5:** SkewIoU is the basis for judging whether the rotated object is detected in the evaluation, so using SkewIoU as the regression loss is more conducive to keeping consistent with the evaluation. This conclusion has been widely recognized in the field of object detection. GWD, KLD and KFIoU (ours) are all designated to better approximate the hard-to-implement SkewIoU loss. Therefore, we need to pay attention to the consistency between the proposed loss and SkewIoU loss in trend level, rather than the value level. In this sense, the size of EMean is not of much significance, and it is mainly used to calculate EVar.
>
>
> > ***Q6: Is the upper bound on the value of KFIoU really important? An estimation of the difference between SkewIoU and KFIoU – if attainable – would be much more valuable.***
>
> **A6:** The calculation of KFIoU simulates the calculation of SkewIoU, so its value domain should be a closed set like IoU, i.e. [0, 1/3], which is necessary for designing an approximate SkewIoU. However, KFIoU only simulates SkewIoU from the trend level, which is sufficient for a loss. Therefore, this paper uses EVar to estimate the difference between the two.

---

> > ### Comment · Reviewer_2M4a · 2022-11-18
> > **Response to author comments**
> >
> > Thank you for your detailed and thorough response. Most of my concerns have been addressed, although I feel it would still be interesting to see the Emean in Table 1. I also think giving an order of magnitude on the variance would help better understand results.
> > After reading through other reviewers comments and your response I maintain my original "weak accept" evaluation of the paper.

---

> > > ### Author Response · Authors · 2022-11-21
> > > **About EMean**
> > >
> > > EMean statistics are shown in the following Table. Compared with the related performance, it is meaningless. As we said before, we pay attention to the consistency between the proposed loss and SkewIoU loss in **trend level**, rather than the **value level**.
> > > For example, the trend between functions $y=x$ and $y=x+10$ is consistent (EMean=10, EVar=0), and they are equivalent as loss functions. In contrast, the $y=2x$ has smaller EMean (<10 within a certain range) with $y=x$, but they are inconsistent (larger EVar).
> > >
> > >
> > > | Loss          | EMean       | EVar            | DOTA-v1.0 | DOTA-v1.5 | DOTA-v2.0 |
> > > | ------------- | ----------- | --------------- | --------- | --------- | --------- |
> > > | Smooth L1     | 0.12901618  | 0.073201718     | 64.17     | 56.10     | 43.06     |
> > > | plain SkewIoU | -           | -               | 68.27     | 59.01     | 45.87     |
> > > | GWD           | -0.38583167 | 0.019041297     | 68.93     | 60.03     | 46.65     |
> > > | KLD           | 0.3291      | 0.007653582     | 71.28     | 62.50     | 47.69     |
> > > | KFIoU         | 0.24926805  | **0.002264243** | **71.60** | **63.75** | **48.94** |

---

> ### Author Response · Authors · 2022-11-10
> **Response to Reviewer 2M4a (Round 1, Part 1/2)**
>
> Thanks for your questions to help improve the paper. We address your concerns and add more results in the main paper and appendix of the uploaded new pdf.
>
> > ***Q1: No indication of uncertainty or context is given on the results. What is the standard deviation? How many runs were performed for each result? Are the baseline results reported or reproduced? If reported, are the settings comparable?***
>
> **A1:** We follow the de-facto protocl in the field of object detection (not specifically rotated object detection) [1-4] that the statistical values like std are not reported and we unfortunately did not record the raw statistics and due to the high volumn of experiments we cannot repdocue the raw numbers during the rebuttal period.
>
>
> In fact, our protocol specifically follows the AlphaRotate open-source to train the model for multiple times (at least three times case by case on specific dataset) and we will release our model and source code to ensure the reproducibility. Basically we find the models are relatively stable -- this is probably the reason why only mean accuracy is reported in this community. The compared methods in the referred paper also only provide the mean value for comparison.
>
> All baselines in Table 1, Table 2 and Table 4, including reported and reproduced, have the same settings because they are all based on the same rotated object detection benchmark AlphaRotate.
>
> [1] Faster r-cnn: Towards real-time object detection with region proposal networks, NeurIPS, 2015
> [2] Mask r-cnn, ICCV, 2017
> [3] Learning high-precision bounding box for rotated object detection via kullback-leibler divergence, NeurIPS, 2021
> [4] Rethinking IoU-based Optimization for Single-stage 3D Object Detection, ECCV, 2022
>
>
> > ***Q2: The center loss is simply added as a correction term. This seems to work well, but it does not really reflect the product of Gaussian at the center of the paper. Have the authors tried using the distance between centers as a multiplicative factor on the KFIoU?***
>
> **A2:** Thanks for your though-provoking suggestion. Here are the results when using the distance between centers as a multiplicative factor on the KFIoU. Note that we have done many experiments for $L_{kf}*L_{c}$ according to different loss weights, and the best results are listed below.
> | Dataset   | Loss           | mAP   |
> | --------- | -------------- | ----- |
> | DOTA-v1.0 | $L_{kf}+L_{c}$ | 70.64 |
> | DOTA-v1.0 | $L_{kf}*L_{c}$ | 63.43 |
>
> We also try to analyze how to approximate SkewIoU without assumptions (center point loss). The main difficulty lies in calculating the area ($S$) of the Gaussian distribution ($\alpha G_{x}(\mu,\Sigma)$, refer to Eq. 5-6 in the main paper) corresponding to the intersection of the two Gaussian distributions.
>
> We assume that there is a plane $l$ on which the maximum projection of $\alpha G_{x}(\mu,\Sigma)$ is the intersection area. First, we set $\alpha G_{x}(\mu,\Sigma)=l$, we have:
>
> $\exp(-\frac{1}{2}(x-\mu)^{\top}\Sigma^{-1}(x-\mu))=\frac{2\pi|\Sigma|^{\frac{1}{2}}}{\alpha}l$
>
> $(x-\mu)^{\top}\Sigma^{-1}(x-\mu)=-2\ln(\frac{2\pi|\Sigma|^{\frac{1}{2}}}{\alpha}l)$
>
> $(x-\mu)^{\top}R\Lambda R^{\top}(x-\mu)=-2\ln(\frac{2\pi|\Sigma|^{\frac{1}{2}}}{\alpha}l)$
>
> The rotation matrix $R$ does not change the size of the intersecting area, so it can be discarded, then we have:
>
> $(x-\mu)^{\top}\Lambda (x-\mu)=-2\ln(\frac{2\pi|\Sigma|^{\frac{1}{2}}}{\alpha}l)$
>
> $\frac{(x-\mu_{x})^2}{\lambda_{1}}+\frac{(y-\mu_{y})^2}{\lambda_{2}}=-2\ln(\frac{2\pi|\Sigma|^{\frac{1}{2}}}{\alpha}l)$
>
> where $\lambda$ represents the eigenvalue.
>
> We find that the above equation has become an ellipse. According to the area formula of the ellipse, we have:
>
> $S=\pi ab$,
> where $a^2=\lambda_{1} (-2\ln(\frac{2\pi|\Sigma|^{\frac{1}{2}}}{\alpha}l))$ and $b^2=\lambda_{2} (-2\ln(\frac{2\pi|\Sigma|^{\frac{1}{2}}}{\alpha}l))$
>
> The key now is the value of $l$, but it seems that $l$ is not a fixed value. The $l$ corresponding to different Gaussian distributions may be different, so the issues seems to be still complicated and requires further research in the future.

---

### Official Review · Reviewer_2Kss · 2022-11-03

**Confidence:** 3
**Correctness:** 4
**Technical Novelty And Significance:** 2
**Empirical Novelty And Significance:** 2
**Recommendation:** 6

**Clarity, Quality, Novelty And Reproducibility:**

The article proposes an extensive review of state-of-the-art approaches for rotated object detection. This gives the reader the context and gives clarity to the paper.

The authors propose an approximation of SkewIoU, called KFIoU. Such term being fully differentiable, it is easier to implement. This answers to a practical problem, thus the novelty is limited.

Reproducibility is ensured by the fact that the authors will make their implementation publicly available.

**Strength And Weaknesses:**

The paper provides an efficient and practical implementation of a loss function for rotated object detection. Such a loss, called KFIoU, does not require any additional hyperparameters and can be easily implemented with fully differentiable functions in existing deep learning frameworks. However, limitations of other implementations are not sufficiently discussed. The authors could insist further on such limitations in order to highlights the advantages brought by the proposed KFIoU.

Some parts can also be rephrased and typos are to be corrected.
In the abstract:
"modeing" -> modeling
"which is used to quickly get the center points between bounding boxes closer to assist the second term" -> it is not clear what you mean by that, especially I don't get what "closer" refers to.
In section 1:
"Rotated object detection is an relatively emerging" -> is a relative ...
"The highlights are as follows" -> the following
2nd point: the first phrase is redundant and refers to highlight number 1.



**Summary Of The Paper:**

The submitted article proposes a practical solution to the rotated object detection problem. To locate arbitrarily oriented objects in images, detectors often rely on the calculation of the SkewIoU. However, its closed-form calculation cannot always be provided and its implementation in popular deep learning framework is limited due to the missing implementation of custom functions. In this context, the authors propose a fully-differentiable approximation of the SkewIoU based on Kalman filter, called KFIoU. Extensive experiments show its practical effectiveness, both in 2-D and 3-D object detection problems.

**Summary Of The Review:**

The paper, although not introducing a new approach to rotated object detection, solves the practical problem of implementation of the SkeweIoU by introducing the KLIoU. Effectiveness of the proposed term is validated by extensive experiments. Although the novelty is limited, the work has a practical impact.

---

> ### Author Response · Authors · 2022-11-10
> **Response to Reviewer 2Kss (Round 1)**
>
> > ***Q1: However, limitations of other implementations are not sufficiently discussed. The authors could insist further on such limitations in order to highlights the advantages brought by the proposed KFIoU.***
>
> **A1:** Thank you for your advice. Here are some supplements:
> - The SkewIoU calculation process is not easy to fulfill due to the complexity of intersection between two rotated boxes[1]. Especially, there exist some custom operations (intersection of two edges and sorting the vertexes etc.) whose derivative functions have not been implemented in the existing deep learning frameworks. See an open-source version with thousands of lines of code for implementing the loss in MMCV: https://github.com/open-mmlab/mmcv/pull/1854. Besides, the calculation of SkewIoU is not differentiable when there are more than eight intersection points between two bounding boxes, i.e. two boundary boxes are completely coincident, or one edge is coincident, which will lead to the failure to obtain very accurate prediction results. Based on the above analysis, developing an easy-to-implement and fully differentiable approximate SkewIoU loss is meaningful and several works [2-6] have been proposed. Note that our new loss only costs tens of lines of code.
>
> - GWD [4] and KLD [5] are relatively advanced approximate SkewIoU losses in recent years, which are easy-to-implement and fully differentiable based on Gaussian modeling. However, they try to approximate SkewIoU loss by specifying a distance which requiry extra hyperparameters tuning and metric selection that vary across datasets and detectors. In contrast, our mechanism level simulation to SkewIoU is more interpretable and natural, and free from hyperparameter tuning.
>
> Table 1 compares the different approximate SkewIoU losses in different aspects, and we also made more detailed supplements at Introduction.
>
>
> > ***Q2: Some parts can also be rephrased and typos are to be corrected. In the abstract: "modeing" -> modeling "which is used to quickly get the center points between bounding boxes closer to assist the second term" -> it is not clear what you mean by that, especially I don't get what "closer" refers to. In section 1: "Rotated object detection is an relatively emerging" -> is a relative ... "The highlights are as follows" -> the following 2nd point: the first phrase is redundant and refers to highlight number 1.***
>
> **A2:** Thank you for your suggestions. We have revised them one by one in the paper.
>
>
> > ***Q3: The authors propose an approximation of SkewIoU, called KFIoU. Such term being fully differentiable, it is easier to implement. This answers to a practical problem, thus the novelty is limited.***
>
> **A3:** We believe that sometimes (if not often) behind a simple yet effective approach, there exists novelty, and our work is such a case.
>
> Recently, many related works [1-6] have been published to propose a practical approximate SkewIoU loss for rotated object detection. Compared with these methods, KFIoU adopts an idea that has never been explored, i.e. Gaussian product, which takes into account the advantages of easy implementation, high performance and free from hyperparameter tuning. **As Reviewer 2M4a said: "KFIoU is an interesting and easy to implement proxy for SkewIoU that follows easy to understand principles". Reviewer Q8Fo said: "The use of Gaussian-modeling-based losses to approximate the SkewIoU loss is not new, but the use of Kalman filters for this use is, and seems to be relevant.". Reviewer jv8Z also thinks KFIoU based on Gaussian product is interesting: "The adoptation of Kalman filter formulation to compute the overlapping in rotated objects is interesting."**
>
> [1] Iou loss for 2d/3d object detection, 3DV, 2019.
> [2] Piou loss: Towards accurate oriented object detection in complex environments, ECCV, 2020.
> [3] Rotation-Robust Intersection over Union for 3D Object Detection, ECCV, 2020.
> [4] Rethinking rotated object detection with gaussian wasserstein distance loss, ICML, 2021.
> [5] Learning high-precision bounding box for rotated object detection via kullback-leibler divergence, NeurIPS, 2021.
> [6] Rethinking IoU-based Optimization for Single-stage 3D Object Detection, ECCV, 2022.

---

> > ### Comment · Reviewer_2Kss · 2022-11-25
> > **Response to authors**
> >
> > In the first submission, from my point of view the paper was not sufficiently clear on the motivations behind the implementation of the proposed loss function. Thus, it was hard to understand the contributions and assess the novelty of the approach, which seemed limited.
> >
> > After rebuttal, the authors have answered to my concerns and modified the paper accordingly to my suggestions.
> >
> > As a consequence, I lean towards acceptance (6+)

---

> > > ### Author Response · Authors · 2022-11-25
> > > **Response to Reviewer 2Kss**
> > >
> > > Thank you. We will continue to improve the paper.

---

### Official Review · Reviewer_cEDF · 2022-11-03

**Confidence:** 4
**Correctness:** 3
**Technical Novelty And Significance:** 3
**Empirical Novelty And Significance:** 2
**Recommendation:** 6

**Clarity, Quality, Novelty And Reproducibility:**

The article can be quite unclear, especially on the motivations behind this method. Neither a theoretical nor an experimental reason is ever clearly given on why the KFIoU loss should be clearly chosen over KLD or GWD loss.

As such, there is only limited novelty in this work compared to previous Gaussian based losses. The comparison of the KFIoU loss to the 3D object detection case is indeed novel; however without comparison to the KLD or GWD losses, this contribution is difficult to judge.

The constant pointing of SkewIoU as being "gradient-training unfriendly" is never truly explained or motivated. Even in the case that SkewIoU is particularly difficult to implement/backpropagate with, the main losses KFIoU is being compared with are losses such as KLD and GWD, and not SkewIoU.

The use of a scatter plot for Fig 3 c is questionable for its clarity. Fig 3 a-b do not seem to showcase a particular improvement of the KFIoU loss over the GWD and KLD losses. It is also unclear why no values of EMean are presented compared to EVar in Table 1. As such, it is difficult to truly conclude on comparisons between GWD KLD and KFIoU losses.

There are many spelling and grammar mistakes in the articles. Some examples are: "Gaussian modeing" (abstract), "works better due
to fully differentiable" (highlights), "are rarely SkewIoU loss" "due to the hard-to-implement
of the SkewIoU" (related work).

**Strength And Weaknesses:**

Strengths: The method proposed is in the logical continuation of Gaussian modeling based losses. Its main advantage is a seemingly closer approximation of the SkewIoU dynamics, while being hyperparameter free.

Weaknesses: The method presented is really similar to the KLD loss, and present only very marginal improvements on DOTA (when using its center loss), and present no comparison with KLD for the other datasets of Table 2 and 3.
The choice of using Kalman Filter (over the previous Gaussian modeling based losses) to create this new loss is never truly motivated. It is not justified why this method should result in a more physically reasonable loss, or one that is theoretically closer to the true SkewIoU value (other than using a similar formula than IoU). The experiments justifying the "alignment" between the losses are not really convincing either (see below).

**Summary Of The Paper:**

This article presents a new loss (KFIoU) approximating the Skew Intersection over Union Loss (SkewIoU) for the rotated object detection problem. This loss is composed of a scale-insensitive center point loss, and a second distance-insensitive term using Gaussian modeling
and Kalman filtering. Compared to other Gaussian modeling based losses like the GWD and KLD losses, this loss is hyperparameter free, and is the first one used in a 3D environment. It is also the closest to the SkewIoU loss, and produce state-of-the-art results.

**Summary Of The Review:**

This article presents a possibly interesting evolution of Gaussian-based SkewIoU approximating losses. KFIoU does not need hyperparameters and has some encouraging experimental results.
However, the method clearly lacks motivation, either theoretically or emperically, to support its lack of novelty compared to previous losses. Furthermore, the paper is often unclear, lacks critical comparisons with KLD and GWD, and with unsubstantiated claims (over SkewIoU "unfriendliness" for example).
For these reasons, I am leaning toward rejection.

---

> ### Author Response · Authors · 2022-11-10
> **Response to Reviewer cEDF (Round 1, Part 3/3)**
>
> > ***Q5: The comparison of the KFIoU loss to the 3D object detection case is indeed novel; however without comparison to the KLD or GWD losses, this contribution is difficult to judge.***
>
> **A5:** The latest experiments of 3D object detection are as follows:
> | 3D Dataset | Method       | 3-D Mod. mAP | BEV Mod. mAP |
> | ---------- | ------------ | ------------ | ------------ |
> | KITTI      | PointPillars | 64.28        | 70.10        |
> | KITTI      | +GWD         | 65.50        | 71.48        |
> | KITTI      | +KLD         | 66.19        | 71.18        |
> | KITTI      | +KFIoU       | **66.71**    | **72.08**    |
>
> We have updated the Table 3 in the latest paper.
>
> > ***Q6: The constant pointing of SkewIoU as being "gradient-training unfriendly" is never truly explained or motivated. Even in the case that SkewIoU is particularly difficult to implement/backpropagate with, the main losses KFIoU is being compared with are losses such as KLD and GWD, and not SkewIoU.***
>
> **A6:** As explained in **A1**, the SkewIoU calculation process is not easy to fulfill due to the complexity of intersection between two rotated boxes [1]. Especially, there exist some custom operations (intersection of two edges and sorting the vertexes etc.) whose derivative functions have not been implemented in the existing deep learning frameworks. See an open-source version with thousands of lines of code for implementing the loss in MMCV: https://github.com/open-mmlab/mmcv/pull/1854. Besides, the calculation of SkewIoU is not differentiable when there are more than eight intersection points between two bounding boxes, i.e. two boundary boxes are completely coincident, or one edge is coincident, which will lead to the failure to obtain very accurate prediction results.
>
> GWD, KLD and our proposed KFIoU all take the approximate SkewIoU loss as the motivation, so we have compared them in Table 1 and Table 4 at the same time.
>
> We have made relevant modifications at Introduction and Contribution, and thank you for your valuable comment.
>
> > ***Q7: The use of a scatter plot for Fig 3c is questionable for its clarity. Fig 3 a-b do not seem to showcase a particular improvement of the KFIoU loss over the GWD and KLD losses.***
>
> **A7:** Figure 3(a-b) mainly reflects the inconsistency between Smooth L1 and SkewIoU.
> The trend consistency (alignment) comparison between GWD/KLD/KFIoU and SkewIoU requires observing scatter distribution in Figure 3c and EVar in Table 1. Specifically, the tighter the scatter distribution is and the closer it is to the linear relationship, the more consistent it is with SkewIoU. At this time, EVar is smaller.
>
> > ***Q8: It is also unclear why no values of EMean are presented compared to EVar in Table 1. As such, it is difficult to truly conclude on comparisons between GWD KLD and KFIoU losses.***
>
> **A8:** SkewIoU is the basis for judging whether the rotated object is detected in the evaluation, so using SkewIoU as the regression loss is more conducive to keeping consistent with the evaluation. This conclusion has been widely recognized in the field of object detection. GWD KLD and KFIoU are all designed to better approximate the hard-to-implement SkewIoU loss. Therefore, we need to pay attention to the consistency between the proposed loss and SkewIoU loss in **trend level**, rather than the **value level**. Therefore, the size of EMean has no reference significance, it mainly used to calculate EVar.
>
> > ***Q9: There are many spelling and grammar mistakes in the articles. Some examples are: "Gaussian modeing" (abstract), "works better due to fully differentiable" (highlights), "are rarely SkewIoU loss" "due to the hard-to-implement of the SkewIoU" (related work).***
>
> **A9:** Thanks for your correction, we have revised them one by one in the paper.
>
>
>
> [1] Iou loss for 2d/3d object detection, 3DV, 2019.
> [2] Piou loss: Towards accurate oriented object detection in complex environments, ECCV, 2020.
> [3] Rotation-Robust Intersection over Union for 3D Object Detection, ECCV, 2020.
> [4] Rethinking IoU-based Optimization for Single-stage 3D Object Detection, ECCV, 2022.
> [5] Generalized Intersection over Union, CVPR, 2019
> [6] Distance-IoU Loss: Faster and Better Learning for Bounding Box Regression, AAAI, 2020.
> [7] Rethinking rotated object detection with gaussian wasserstein distance loss, ICML, 2021.
> [8] Learning high-precision bounding box for rotated object detection via kullback-leibler divergence, NeurIPS, 2021.

---

> > ### Author Response · Authors · 2022-11-21
> > **Q8/A8 update**
> >
> > EMean statistics are shown in the following Table. Compared with the related performance, it is meaningless. As we said before, we pay attention to the consistency between the proposed loss and SkewIoU loss in **trend level**, rather than the **value level**.
> > For example, the trend between functions $y=x$ and $y=x+10$ is consistent (EMean=10, EVar=0), and they are equivalent as loss functions. In contrast, the $y=2x$ has smaller EMean (<10 within a certain range) with $y=x$, but they are inconsistent (larger EVar).
> >
> >
> > | Loss          | EMean       | EVar            | DOTA-v1.0 | DOTA-v1.5 | DOTA-v2.0 |
> > | ------------- | ----------- | --------------- | --------- | --------- | --------- |
> > | Smooth L1     | 0.12901618  | 0.073201718     | 64.17     | 56.10     | 43.06     |
> > | plain SkewIoU | -           | -               | 68.27     | 59.01     | 45.87     |
> > | GWD           | -0.38583167 | 0.019041297     | 68.93     | 60.03     | 46.65     |
> > | KLD           | 0.3291      | 0.007653582     | 71.28     | 62.50     | 47.69     |
> > | KFIoU         | 0.24926805  | **0.002264243** | **71.60** | **63.75** | **48.94** |

---

> > > ### Comment · Reviewer_cEDF · 2022-11-25
> > > **Response to authors**
> > >
> > > I thank the authors for their responses. They have answered well many of my concerns on their previous submission.
> > >
> > > Results are now sufficiently compared with KLD and GWD. The improvements are limited but are indeed consistent between the different datasets. The justification for preferring to showcase EVar is indeed valid but I thank the authors for giving the values of EMean.
> > > The clarity and motivation of their submission has also improved.
> > >
> > > The link with Kalman filter was fragile, and the new repositioning of their main contributions is clearer. Their claim of simplicity is valid but their point of differentiability is still a bit weak since the only problems are differentiable IoU are edge cases.
> > >
> > > The novelty of their method is still relatively weak, with marginal empirical improvements compared to other Gaussian based losses. However it is consistent between datasets, and better motivated than before. The writing and clarity of the submission has also improved (although some errors remain ("due to it is")). Therefore, and also based on the other reviewers' comments I will update my rating to a weak accept of 6.

---

> > > > ### Author Response · Authors · 2022-11-25
> > > > **Response to Reviewer cEDF**
> > > >
> > > > Thank you for your valuable suggestions and we will continue to improve our paper.

---

> ### Author Response · Authors · 2022-11-10
> **Response to Reviewer cEDF (Round 1, Part 2/3)**
>
> > ***Q3: The article can be quite unclear, especially on the motivations behind this method. Neither a theoretical nor an experimental reason is ever clearly given on why the KFIoU loss should be clearly chosen over KLD or GWD loss.***
>
> **A3:** Thanks for your comments to remind us to improve the presentation. We seize this open review discussion opportunity to clarify the points and have also made revisions in the Introduction and Contribution parts of the main pdf.
>
>
> The inconsistency between regression loss and evaluation (i.e. mAP, largely depending on SkewIoU) is an important issue in the field of rotated object detection, which is also the background introduced at the beginning of the Introduction.
>
> The most direct way to solve this issue is to use IoU loss. However, for rotated object detection, the SkewIoU calculation process is technically nontrivial due to the complexity of intersection between two rotated boxes [1]. Practically, there exist some custom operations (intersection of two edges and sorting the vertexes etc.) whose complex derivative functions have not been implemented in the existing deep learning frameworks. See an open-source version with thousands of lines of code for implementing the loss in MMCV: https://github.com/open-mmlab/mmcv/pull/1854.
>
> Besides, the calculation of SkewIoU is not differentiable when there are more than eight intersection points between two bounding boxes, i.e. two boundary boxes are completely coincident, or one edge is coincident, which will lead to the failure to obtain very accurate prediction results. Based on the above analysis, developing an easy-to-implement and fully differentiable approximate SkewIoU loss is meaningful and several works [2-4,7-8], including GWD [7], KLD [8] have been proposed.
>
> GWD and KLD use distribution distance to approximate SkewIoU, which is still not consistent with the evaluation, and additional hyperparameters are introduced. In contrast, KFIoU use similarity metirc to approximate SkewIoU, achieving a more consistent change trend with the evaluation (i.e. mAP, largely depending on SkewIoU) measured by proposed EVar, while free from hyperparameters tuning. Table 1 summaries the properties and performance of different regression losses.
>
>
> > ***Q4: As such, there is only limited novelty in this work compared to previous Gaussian based losses.***
>
> **A4:** Compared with other peer methods [1-4, 7-8], KFIoU adopts an idea that has never been explored, i.e. Gaussian product, which takes into account the advantages of straightforward implementation, strong performance and free from hyperparameter tuning.
>
> There are some side evidences to support our claim: **Reviewer 2M4a said: "KFIoU is an interesting and easy to implement proxy for SkewIoU that follows easy to understand principles". Reviewer Q8Fo said: "The use of Gaussian-modeling-based losses to approximate the SkewIoU loss is not new, but the use of Kalman filters for this use is, and seems to be relevant.". Reviewer jv8Z also thinks KFIoU based on Gaussian product is interesting: "The adoptation of Kalman filter formulation to compute the overlapping in rotated objects is interesting."**
>
>
> [1] Iou loss for 2d/3d object detection, 3DV, 2019.
> [2] Piou loss: Towards accurate oriented object detection in complex environments, ECCV, 2020.
> [3] Rotation-Robust Intersection over Union for 3D Object Detection, ECCV, 2020.
> [4] Rethinking IoU-based Optimization for Single-stage 3D Object Detection, ECCV, 2022.
> [5] Generalized Intersection over Union, CVPR, 2019
> [6] Distance-IoU Loss: Faster and Better Learning for Bounding Box Regression, AAAI, 2020.
> [7] Rethinking rotated object detection with gaussian wasserstein distance loss, ICML, 2021.
> [8] Learning high-precision bounding box for rotated object detection via kullback-leibler divergence, NeurIPS, 2021.

---

> ### Author Response · Authors · 2022-11-10
> **Response to Reviewer cEDF (Round 1, Part 1/3)**
>
> Thanks for your careful reading of our paper, and giving important suggestions for us to improve the paper. We hope our response and clarifiction can ease some of your concerns and you could reconsider your rating. We are happy to answer your any further questions.
>
> > ***Q1: The method presented is really similar to the KLD loss, and present only very marginal improvements on DOTA (when using its center loss).***
>
> **A1:** Thank you for your comments. Though the center point term KLD will be used by KFIoU, there are still fundamental differences between KFIoU and KLD losses.
>
> - First, the center point loss ($L_{c}$) is the starting point assumption of $L_{kf}$, so it is not the main contribution of this paper. The core novelty of this paper is the Gaussian product, i.e. the second item $L_{kf}$.
> - Secondly, the core of KLD is the distribution distance metric (KL divergence), while KFIoU is the similarity metirc (approximate SkewIoU). We believe the latter is more reasonable for resolving the inconsistency between regression loss and evaluation (i.e. mAP, largely depending on SkewIoU). This is because IoU/SkewIoU loss is (currently) the most effective way to solve the inconsistency problem of regression loss and evaluation, which is widely recognized in the field of horizontal/rotated object detection [1-6].
> - Third, the distribution distance metric also incurs  hyperparameters to tune for KLD while our method is hyperparameter-free.
> - Fourth, KFIoU exceeds KLD 0.32%, 1.25% and 1.25% on DOTA-v1.0, DOTA-v1.5 and DOTA-v2.0 in Table 1, respectively. We believe this is a significant gain for this competitive area, especially in the more challenging DOTA-v1.5 and DOTA-v2.0, which contain more tiny objects (less than 10 pixels).
>
> In general, KFIoU has the advantage over KLD in that it can better optimize the three parameters of ($w, h, \theta$), that is, get better performance while free from hyperparameter tuning.
>
>
> > ***Q2: Present no comparison with KLD for the other datasets of Table 2 and 3.***
>
> **A2:** Thank you for your valuable suggestion. We will try our best to supplement more experiments compared with KLD. The following is a part of the current supplement (being continuously increased):
>
> | 2D Dataset | Loss      | Hmean     |
> | ---------- | --------- | --------- |
> | ICDAR2015  | Smooth L1 | 69.78     |
> | ICDAR2015  | GWD       | 74.29     |
> | ICDAR2015  | KLD       | 75.32     |
> | ICDAR2015  | KFIoU     | **75.90** |
>
> | 2D Dataset | Loss      | Car       | Plane     | mAP       |
> | ---------- | --------- | --------- | --------- | --------- |
> | UCAS-AOD   | Smooth L1 | 92.62     | 96.50     | 94.56     |
> | UCAS-AOD   | GWD       | 94.03     | 96.86     | 95.44     |
> | UCAS-AOD   | KLD       | 94.34     | 97.94     | 96.14     |
> | UCAS-AOD   | KFIoU     | **94.51** | **98.41** | **96.46** |
>
> | Loss      | SSDD inshore | HRSID inshore |
> | --------- | ------------ | ------------- |
> | Smooth L1 | 68.47        | 51.41         |
> | GWD       | 77.71        | 51.11         |
> | KLD       | 76.84        | 52.80         |
> | KFIoU     | **77.89**    | **53.45**     |
>
> | 3D Dataset | Method       | 3-D Mod. mAP | BEV Mod. mAP |
> | ---------- | ------------ | ------------ | ------------ |
> | KITTI      | PointPillars | 64.28        | 70.10        |
> | KITTI      | +GWD         | 65.50        | 71.48        |
> | KITTI      | +KLD         | 66.19        | 71.18        |
> | KITTI      | +KFIoU       | **66.71**    | **72.08**    |
>
> [1] Iou loss for 2d/3d object detection, 3DV, 2019.
> [2] Piou loss: Towards accurate oriented object detection in complex environments, ECCV, 2020.
> [3] Rotation-Robust Intersection over Union for 3D Object Detection, ECCV, 2020.
> [4] Rethinking IoU-based Optimization for Single-stage 3D Object Detection, ECCV, 2022.
> [5] Generalized Intersection over Union, CVPR, 2019
> [6] Distance-IoU Loss: Faster and Better Learning for Bounding Box Regression, AAAI, 2020.
> [7] Rethinking rotated object detection with gaussian wasserstein distance loss, ICML, 2021.
> [8] Learning high-precision bounding box for rotated object detection via kullback-leibler divergence, NeurIPS, 2021.

---

### Author Response · Authors · 2022-11-18
**Comments to everyone.**

Dear Reviewers,

Approaching the pdf updating ddl, is there anything needing added.

Best,

Paper993 Authors

---

### Decision · Program_Chairs · 2023-01-20

**Decision:**

Accept: poster

**Justification For Why Not Higher Score:**

- The original SkewIoU loss is non-differentiable only in corner case situations
- Limited novelty with respect to existing works on Gaussian modelling (GWD and KLD)
- Small experimental gains (although consistent)

**Justification For Why Not Lower Score:**

- Sound approach and paper clearly presented. The approach can be a reference baseline for rotated object detection
- Hyper-parameter free and thus easy to tune approach
- Huge and thorough experimental evaluation, in 2D and 3D, with consistent gains

**Metareview: Summary, Strengths And Weaknesses:**

This paper introduces the KFIoU loss for rotated object detection. KFIoU approximates the Skew Intersection over Union Loss (SkewIoU) with a Gaussian modelling, making the loss differentiable, and is hyper-parameters free compared to recent works from the literature, e.g. GWD and KLD. KFIoU also includes a center point loss to take into account distance between rotated bounding boxes. Experiments are conducted on aerial images, faces, scene text datasets and autonomous driving benchmarks (KITTI).
The paper initially received mixed reviews, with one clear acceptance recommendation, two borderline accept, one borderline reject, and one reject recommendation. The main concerns pointed out by reviewers related to the motivation of the approach, its positioning with respect to KLD/GWD, its over-claimed connection to Kalman filtering, experimental validation (significance of the results, comparison between KFIoU limited to DOTA).
The rebuttal and paper's update did a good job in answering reviewers' concerns by adding new conclusive experiments and clarifying the paper's motivations. After discussions, the was a consensus among reviewers that the paper should be accepted.

The AC carefully reads the submission. The AC considers that the approach is incremental yet interesting and that the last version of the paper is clearly written. Although the justification for not directly optimizing over SkewIoU might be questionable (since the original loss is non-differentiable only in corner case situations), the proposed smooth surrogate loss is sound, hyper-parameter free and thus easy to tune. The experiments are thorough and have been conducted on 2D but also on 3D benchmarks, showing small but consistent gains of KFIoU against the latest baselines.
Therefore, the AC recommends paper acceptance.

**Note From Pc:**

if the above contains the word "oral" or "spotlight" please see: "oral" presentation means -> notable-top-5% and "spotlight" means -> notable-top-25%. As stated in our emails, we are disassociating presentation type from AC recommendations